# Effects of atmospheric $CO_2$ variability of the past 800 ka on the
biomes of Southeast Africa
Lydie M. Dupont, MARUM - Center for Marine Environmental Sciences, University of
Bremen
Thibaut Caley - EPOC, UMR 5805, CNRS, University of Bordeaux, Pessac, France
Isla S. Castañeda - University of Massachusetts Amherst, Department of Geosciences,
Amherst, MA 01003 USA
## Abstract
Very little is known about the impact of atmospheric carbon dioxide pressure ($p$CO$_2$) on the shaping
of biomes. The development of $p$CO$_2$ throughout the Brunhes Chron may be considered a natural
experiment to elucidate relationships between vegetation and $p$CO$_2$. While the glacial periods show
low to very low values (~220 to ~190 ppmv, respectively), the $p$CO$_2$ levels of the interglacial periods
vary from intermediate to relatively high (~250 to more than 270, respectively). To study the
influence of $p$CO$_2$ on the Pleistocene development of SE African vegetation, we used the pollen
record of a marine core (MD96-2048) retrieved from Delagoa Bight south of the Limpopo River
mouth in combination with stable isotopes and geochemical proxies. Applying endmember analysis,
four pollen assemblages could be distinguished representing different biomes: heathland, mountain
forest, shrubland and woodland. We find that the vegetation of the Limpopo River catchment and
the coastal region of southern Mozambique is not only influenced by hydroclimate but also by
temperature and atmospheric $p$CO$_2$. Our results suggest that the extension of mountain forest
occurred during those parts of the glacials when $p$CO$_2$ and temperatures were moderate and that
only during the colder periods when atmospheric $p$CO$_2$ was low (less than 220 ppmv) open
ericaceous vegetation including C4 sedges extended. The main development of woodlands in the
area took place after the Mid-Brunhes Event (~430 ka) when interglacial $p$CO$_2$ levels regularly rose
over 270 ppmv.
### *Keywords*
*Palynology, Pleistocene, Limpopo River catchment, atmospheric carbon dioxide, Mid-Brunhes Event*
## Short summary
Multiproxy study of marine sediments off the Limpopo River mouth spanning the Late Pleistocene
reveals the impact of atmospheric carbon dioxide on the development of the vegetation of Southeast
Africa and indicates changes in the interglacial vegetation before and after the Mid-Brunhes Event.
## Highlight
A multiproxy study of marine sediments off the Limpopo River mouth spanning the Late Pleistocene
reveals the impact of atmospheric carbon dioxide on the development of the vegetation of Southeast
Africa and indicates changes in the interglacial vegetation before and after the Mid-Brunhes Event.
The unique record of detailed vegetation change in SE Africa over the entire Brunhes Chron
demonstrates the expansion of glacial vegetation in southern Africa when atmospheric CO$_2$
concentration was low and the development of miombo woodland in SE Africa during successive
interglacials after the Mid-Brunhes Transition.

# 1 Introduction

Understanding the role of atmospheric carbon dioxide pressure ($p$CO$_2$) is paramount for the interpretation of the of the paleo-vegetation record. The effects of low $p$CO$_2$ on glacial vegetation have been discussed in a number of studies [Ehleringer et al. 1997, Jolly & Haxeltine 1997, Cowling & Sykes 1999, Prentice & Harrison 2009, Prentice et al. 2017] predicting that glacial increases in C4 vegetation favored by low atmospheric CO$_2$ would have opened the landscape and lowered the tree line. Comparing records of this glacial C4-rich vegetation with modern analogues could have led to estimating more severe aridity than actually occurred during the Last Glacial Maximum. These studies [opt cit.], however, mostly cover the last glacial-interglacial transition and do not examine periods with intermediate $p$CO$_2$ such as during the Early Glacial (MIS 5a-d) or interglacials prior to 430 thousand years ago (430 ka). Comparing the vegetation record of subsequent climate cycles showing different CO$_2$ levels might help to better understand the effects of $p$CO$_2$ on the vegetation.

During the Brunhes Chron (past 780 ka) the length of the glacial cycles became much longer lasting roughly 100 ka due to a strong non-linear response of the ice sheets to solar forcing [Mudelsee & Stattegger 1997]. Model experiments of Ganopolski & Calov [2011] indicate that low atmospheric CO$_2$ concentrations are a prerequisite for the long duration of the glacial cycles of the past 800 ka. Then, roughly mid-way through the Brunhes Chron, the amplitude of the climate cycles shifted with a change in the maximum CO$_2$ concentration during interglacials.

This so-called Mid-Brunhes Event (MBE) [Jansen et al. 1986] - also called Mid-Brunhes Transition - occurred about 430 ka ago and marks the transition between interglacials characterized by rather low atmospheric CO$_2$ around 240 ppm (parts of Marine Isotope Stages (MIS) 19, 17, 15, 13) to interglacials in which CO$_2$ levels reached 270 ppmv or more (parts of MIS 11, 9, 7, 5, 1) [Lüthi et al. 2008, Bereiter et al. 2015]. The climate transition of the MBE has been extensively studied using Earth System Models of Intermediate Complexity. Yin & Berger [2010] stress the importance of forcing by austral summer insolation and Yin & Berger [2012] argue that the model vegetation (tree-fraction) was forced by precession through precipitation at low latitudes. Both papers show the necessity to include the change in atmospheric CO$_2$ in the explanation of the MBE [Yin & Berger, 2010, 2012]. Yin [2013], however, concludes that it is not necessary to invoke a sudden event around 430 ka to explain the increased interglacial CO$_2$; the differences between interglacials before and after the MBE can be explained by individual responses in Southern Ocean ventilation and deep-sea temperature to various combinations of the astronomical parameters. On the other hand, statistical analysis suggests a dominant role of the carbon cycle, which changed over the MBE [Barth et al. 2018]. Paillard [2017] developed a conceptual model of orbital forcing of the carbon cycle in which sea-level fluctuations and the effects on carbon burial are decisive during shifts in the climate system. Further modelling by Bouttes et al. [2018] showed qualitative agreement with the paleodata of pre- and post-MBE interglacials but largely underestimated the amplitude of the changes. Moreover, the simulated vegetation seems to counteract the effects of the oceanic response [Bouttes et al. 2018]. Thus the vegetation, in particular at low latitudes, may play a crucial but poorly understood role in the climate system.

Comparing records of pre- and post-MBE interglacials could offer insight in the interglacial climate at
different levels of $CO_2$ [Foley et al. 1994, Swann et al. 2010]. We define interglacials after PAGES
[2016] listing MIS 19c, 17c, 15a, 15e, 13a as pre-MBE and MIS 11c, 9e, 7e, 7a-c, 5e, 1 as post-MBE.
Currently, only a handful of vegetation records covering the entire Brunhes Chron have sufficient
temporal resolution to enable comparisons between interglacials before and after the Mid-Brunhes
transition. These records are from the eastern Mediterranean, the Colombian Andes [PAGES 2016],

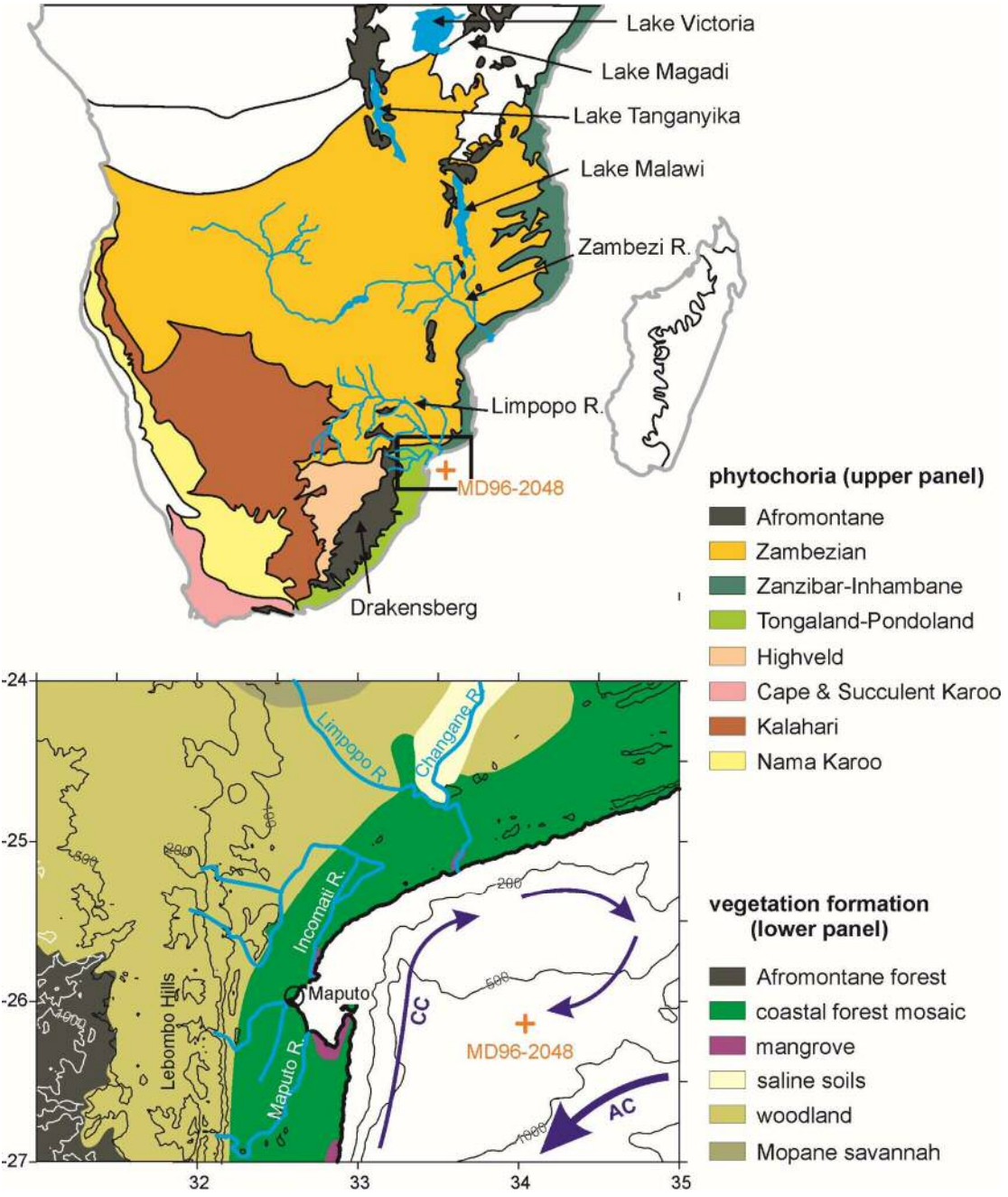

Figure 1. Upper panel: map of southern Africa with the main phytochoria after White (1983). Lower panel: site location of MD96-2048; main vegetation formations; main rivers; 100 m, 200 m, 500 m, and 1000m contours; 200 m, 500 m, and 1000m bathymetric contours; Agulhas (AC) and counter currents (CC) forming a coastal Delagoa Bight Lee Eddy. Zambezian vegetation woodland and savannah north of ~25°30'S, Tongaland-Pondoland coastal forests south of ~25°30'S, Zanzibar-Inhambane coastal forests east of 33–34°E. West of the escarpment with Afromontane forest rises the interior plateau covered with Highveld grasslands rises.

West and East Africa [Dupont et al. 1989, Miller & Goslin 2014, Castañeda et al. 2016, Johnson et al. 2016, Ivory et al. 2018, Owen et al. 2018]. The Andean pollen record is strongly influenced by the immigration of oak from North America during MIS 12 [Torres et al. 2013]. For the eastern Mediterranean a decline in plant diversity is observed at Tenaghi Phillipon (Greece) where the modern Mediterranean oak forests gradual emerged in the interglacials after MIS 16 but before the MBE [Tzedakis et al. 2006, 2009]. The West African record of Lake Bosumtwi in Ghana allows identification of six forest assemblages since 540 ka related to the interglacials of MIS 13, 11, 9, 7, 5e, and 1. The forests assemblage of MIS 13, however, does not show a strong contrast with those of the interglacials after the MBE [Miller & Goslin 2014]. The marine pollen record of ODP Site 658 off Cape Blanc tracks the latitudinal position of the open grass-rich vegetation zones at the boundary between Sahara and Sahel suggesting shifting vegetation zones between glacials and interglacials [Dupont & Hooghiemstra 1989, Dupont et al. 1989]. The drier interglacials occurred after MIS 9, which indicates a transition after the MBE to more arid conditions.  Additionally, stable carbon isotope records from Chinese loess sections indicate interglacial-glacial variability in the C3-C4 proportions of the vegetation [Lyu et al. 2018, Sun et al. 2019]. However, the latter records do not show a prominent vegetation shift over the MBE.

For East Africa two terrestrial records and a marine one are available. From Lake Malawi, Johnson et al. [2016] infer wetter conditions and increased woodland vegetation between 800 and 400 ka based on the stable carbon isotopic composition of plant wax shifting from less to more strongly depleted values. Also from Lake Malawi, Ivory et al. [2018] published a pollen record of the past 600 ka revealing a number of phases of miombo woodland and mountain forest alternating with savannah vegetation (dry woodland and wooded grassland). Recently, a new record from Lake Magadi (Kenya) has been published indicating a change from wetter conditions to more aridity after 500 ka contrasting the Lake Malawi results [Owen et al. 2018]. In Lake Magadi, the representation of *Podocarpus* decreased over the MBE, while open grassy vegetation and salinity of the lake increased [Owen et al. 2018]. Neither the Lake Malawi nor the Lake Magadi records show dominant interglacial-glacial variability.

The marine record retrieved south of the Limpopo River mouth (Core MD96-2048) allows inferences about vegetation and climate in the catchment area of the Limpopo River draining large areas of South Africa, Botswana, Zimbabwe and Mozambique. Based on sediment chemistry, Caley et al. [2018] reported the effects of increased summer insolation in increased fluvial discharge and variability associated with eccentricity, which modulates precession amplitudes. Superimposed on the orbital-scale precipitation variability, a long-term trend from 1000 to 600 ka towards increased aridity in southeastern Africa was found [Caley et al. 2018]. The plant leaf wax carbon isotopic (hereafter $\delta^{13}C_{wax}$) record of the same core was originally interpreted as reflecting a trend toward increasingly drier glacials and wetter interglacials over the past 800 ka [Castañeda et al. 2016]. Additionally, the average chain lengths of the plant leaf waxes exhibit a stepwise decrease at 430 ka suggesting a change from more shrub vegetation before the MBE to a larger contribution of trees during the post-MBE interglacials [Castañeda et al. 2016]. Thus, a pollen record of MD96-2048 has the potential to register the changes in interglacial vegetation cover over the MBE. We might expect a change of Southern Hemisphere vegetation being less ambiguous than the changes found on the Northern Hemisphere (see above), because modelling indicates that the effects of the MBE were more pronounced on the Southern Hemisphere [Yin & Berger 2010]. Until now, the palynology of only the last 350 ka has been published [Dupont et al. 2011] and, therefore, here we extend the pollen record of MD96-2048 to cover the past 800 ka in sufficient resolution. As described below, our new palynology results have led to the re-interpretation of the MD96-2048 $\delta^{13}C_{wax}$ record [Castañeda et al. 2016].

## 1  Previous work on Core MD96-2048

The sediments of MD96-2048 were retrieved in the middle of the Delagoa Bight (Figure 1) from the southern Limpopo cone forming a depot center that has been build up at least since the Late Miocene [Martin 1981]. The site collects terrestrial material including pollen and spores mostly from the rivers that discharge into the Delagoa Bight of which the Limpopo River is the biggest draining large areas of northern South Africa and southern Mozambique. Apart from the offshore winds descending from the interior plateau, so-called Bergwinds, the predominant wind direction is landward [Tyson & Preston-Whyte 2000] and aeolian input of terrestrial material is probably minor. Thus, pollen source areas would cover the region north of the Delagoa Bight in southern Mozambique and the region west of Delagoa of the Lebombo hills and the Drakensberg Escarpment [Dupont et al. 2011].

A wide variety of measurements have been performed on MD96-2048 sediments. Caley et al. [2011, 2018] recorded stable oxygen isotopes of benthic foraminifers (*Planulina wuellerstorfi*) providing a stable oxygen stratigraphy and age model aligned to the global stack LR04 [Lisiecki & Raymo 2005] for the past 2200 ka. Trace element (Mg/Ca ratios) of the planktic foraminifer *Globigernoides ruber sensu stricto* and foraminifer assemblages were combined to produce a robust sea surface temperature (SST) record [Caley et al., 2018]. High resolution (0.5 cm) XRF-scanning has been performed over the total core length, of which the iron-calcium ratios, ln(Fe/Ca), were used to estimate fluvial terrestrial input variability [Caley et al., 2018]. At millennial resolution, higher plant leaf wax (*n*-alkane) concentrations and ratios and compound specific stable carbon isotopes ($\delta^{13}C_{wax}$) provided a record of vegetation changes in terms of open versus closed canopy and C4 versus C3 plants of the past 800 ka [Castañeda et al. 2016]. A very low resolution leaf wax deuterium isotopic record was generated [Caley et al. 2018], and in conjunction with other high-resolution proxies including ln(Fe/Ca), was used to reconstruct rainfall and Limpopo River runoff during the past 2.0 Ma.

## Present-day climate and vegetation

Modern climate is seasonal with the rainy season in summer (November to March). Yearly precipitation ranges from 600 mm in the lowlands to 1400 mm in the mountains, whereby rains are more frequent along the coast under the influence of SSTs [Jury et al. 1993, Reason & Mulenga 1999]. Annual average temperatures range from 24 to 16°C but in the highlands clear winter nights may be frosty.

The modern vegetation of this area belongs to the forest, Highveld grassland, and savannah biomes and also includes azonal vegetation (Figure 1) [Dupont et al. 2011 and references therein]. The natural potential vegetation of the coastal belt is forest, although at present it is almost gone; north of the Limpopo River mouth rain forests belong to the Inhambane phytogeographical mosaic and south of the Limpopo River the forest belongs to the Tongaland-Pondoland regional mosaic [White 1983]. The vegetation of the northern part of the Tongaland-Pondoland region is the Northern Coastal Forest [Mucina & Rutherford 2006]. Semi-deciduous forest is found in the Lebombo hills [Kersberg 1996]. Afromontane forest and Highveld grasslands grow along the escarpment and on the mountains. The savannahs of the Zambezian phytogeographical region including e.g. the miombo dry forest occur further inland [White 1983]. Azonal vegetation consists of freshwater swamps, alluvial, and seashore vegetation and mangroves [Mucina & Rutherford 2006].

# Material and Methods

Pollen analysis of the 37.59 m long core MD96-2048 (26°10'S 34°01'E, 660m water depth) was extended with 65 samples down core to 12 m (790 ka). Average sampling distance for the Brunhes part was 7 cm reaching an average temporal resolution of 4 ka according to the age model based on

the stable oxygen isotope stratigraphy of benthic foraminifers [Caley et al. 2011, 2018]. Two older
windows have been sampled; 20 samples between 15 and 26 m (943-1537 ka) and 19 samples
between 30 and 36 m (1785-2143 ka) were taken with an average resolution of 31 and 20 ka,
respectively.
Pollen preparation has been described in [Dupont et al. 2011]. In summary, samples were decalcified
with HCl (~10%), treated with HF (~40%) for two days, ultrasonically sieved over an 8-µm screen and,
if necessary, decanted. The samples were spiked with two *Lycopodium* spore tablets (either of batch
#938934 or batch #177745). Residues were mounted in glycerol and pollen and spores examined at
400x or 1000x magnification. Percentages are expressed based on the total of pollen and spores
ranging from over 400 to 60 – only in six samples this sum amounts to less than 100. Confidence
intervals (95%) were calculated after Maher [1972, 1981]. Pollen have been identified using the
reference collection of African pollen grains of the Department of Palynology and Climate Dynamics

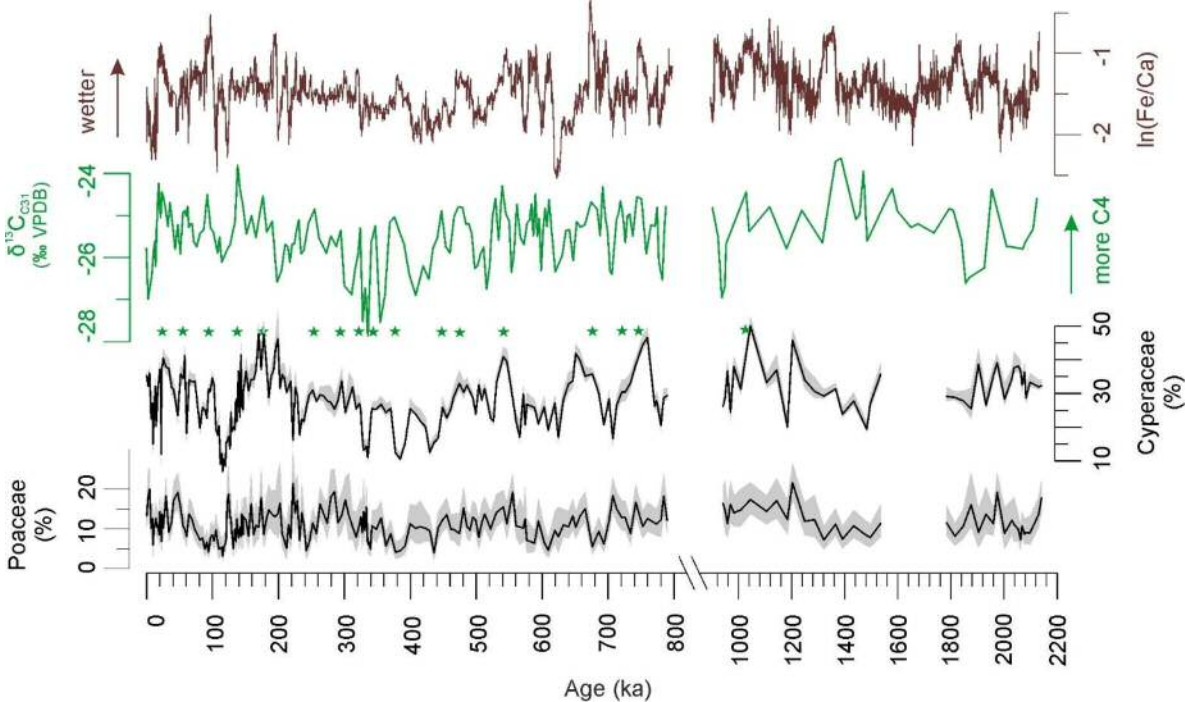

Figure 2. Indicators of C4 vegetation and terrestrial input. From top to bottom: elmental Fe/Ca ratios [Caley et al. 2018]; less negative values indicate relatively wetter conditions, $\delta^{13}C_{wax}$ of the *n*-alkane $C_{31}$ [Castañeda et al. 2016, Caley et al. 2018]; less negative values of around -24‰ indicae more C4 inputs while more negative values of around -28‰ indicate more C3 inputs, Cyperaceae (sedges) pollen percentages [Dupont et al., 2011, this study], Poaceae (grass) pollen percentages [Dupont et al., 2011, this study]. Shaded areas denote 95% confidence intervals after [Maher 1972]. Stars denote corresponding maxima in Cyperaceae pollen percentages and the stable carbon isotopes indicating C4 vegetation. VPDB: Vienna Pee Dee Belemnite. Note the scale break.

of the University of Göttingen, the African Pollen Database collection, and literature [e.g. Bonnefille
& Riollet 1980, Scott 1982, Köhler & Brückner 1982, 1989, Schüler & Hemp 2016].
We assigned pollen taxa to groups such as riparian, woodland, forest, etc. (Supplementary Table 1)
using information given by Scott [1982], White [1983], Beentje [1994], Kersberg [1996], Coates-
Palgrave [2002], Vincens et al. [2007]. Additionally, we carried out a multivariate analysis in the form
of an endmember model unmixing procedure [Weltje, 1997], the statistics of which are specifically
designed for the treatment of percentage data. We regard the pollen percentages as a series of
pollen assemblage mixtures, whereby each modelled endmember may be interpreted as the
representation of one or more biomes. This linear mixing model can be compared to a ternary
diagram but allowing for more than three axes. We use a version of the unmixer algorithm
programmed in MATLAB by Dave Heslop in 2008. Taxa occurring in 6 or more samples (listed in
Supplementary Table 2) were used in the endmember modelling (148 of 231 taxa in 220 samples).
We selected a model with four components explaining more than 95% of the variance ($r^2$ =0.953).
Iteration was stopped at 1000x resulting in a convexity at termination of -1.6881. Significance level at
99% for taxa to score on the assemblages was 0.018.
To study the correlations between different parameters, we used a linear regression model (least
square regression) on linearly interpolated values (5 ka steps) from 0 to 790 ka. Correlation
coefficients are given in Table 1. For interpolation and testing the correlation, we used the package
PAST [Hammer et al. 2001].

# Result and Discussion

## Terrestrial input and provenance of the C4 plant wax

Pollen percentages of Cyperaceae (sedges) and Poaceae (grasses) are plotted in Figure 2 together
with the $\delta^{13}C_{wax}$ of the $C_{31}$ $n$-alkane and XRF-scanning data, ln(Fe/Ca), the natural logarithm of
elemental ratios of iron over calcium. Comparing the records of Cyperaceae and $\delta^{13}C_{wax}$ reveals that
high relative amounts of C4 plant material co-varied with increased representation of sedges. They
also co-varied with higher terrestrial input indicated by ln(Fe/Ca), and increased precipitation as
suggested by deuterium of the $C_{31}$ $n$-alkane [Caley et al., 2018]. We substantiated the correlations for
the Brunhes Chron between pollen percentages, leaf waxes and elemental ratios in Table 1. Leaf wax
data are after Castañeda et al. [2016] including Average Chain Length (ACL) of the $C_{27}$ - $C_{33}$$n$-alkanes,
the ratio of $C_{31}/(C_{31}+C_{29})$ and $\delta^{13}C_{wax}$. XRF ln(Fe/Ca) ratios are from Caley et al. [2018]. Significant
correlation is found between the leaf wax parameters and Cyperaceae data but not between $\delta^{13}C_{wax}$
(indicative of C4 inputs) and Poaceae pollen percentages - although a correlation exists between
Cyperaceae and Poaceae percentages. While the sedge pollen percentages fluctuate between 10 and
50% (mostly > 20%), the percentages of grass pollen are always lower than 20%. Such low grass
pollen values have not been found adjacent C4 grass dominated biomes (mainly savannahs) on the
western side of the continent [Dupont 2011]. It is, therefore, likely that in sediments of MD96-2048
the C4 component of the plant wax originated from C4 sedges rather than from C4 grasses.
South Africa has 68 species of Cyperaceae (sedges) of which 28 use the C4 pathway (among the 10
*Cyperus* species 8 are C4) predominantly growing in the northern part of the country [Stock et al.
2004]. They are an important constituent of tropical swamps and riversides [Chapman et al. 2001].

Table 1. Correlation coeffients calculated with PAST [Hammer et al 2001]. Significant correlations are underlined (95%) or bold and underlined (99%). Average Chain Length (ACL), ratio of concentrations of $C_{31}/($ $C_{31} + C_{29})$, and stable carbon isotope composition of the $C_{31}$$n$-alkane ($\delta^{13}C_{wax}$) after Castañeda et al. [2016]. Cyperaceae and Poaceae pollen percentages (of total pollen and spores) after Dupont et al. [2011] and this study. ln(Fe/Ca) data after Caley et al. [2018].

| $r^2$ | ACL | Ratio $C_{31}/(C_{29}+C_{31})$ | $\delta^{13}C_{wax}$ (‰) | Cyperaceae (%) | Poaceae (%) | XRF ln (Fe/Ca) |
|---|---|---|---|---|---|---|
| ACL | 1 | | | | | |
| Ratio $C_{31}/(C_{31}+C_{29})$ | **0.635** | 1 | | | | |
| $\delta^{13}C_{wax}$ | **0.079** | **0.180** | 1 | | | |
| Cyperaceae (%) | 0.027 | **0.140** | **0.142** | 1 | | |
| Poaceae (%) | 0.003 | 0.016 | 0.003 | **0.165** | 1 | |
| XRF ln(Fe/Ca) | 0.016 | 0.032 | **0.227** | **0.110** | 0.011 | 1 |

An inventory of six modern wetlands between 500 and 1900m in KwaZulu Natal shows that C4 grasses dominate the dry surroundings of the wetlands at all altitudes [Kotze & O'Connor 2000]. In the wet parts of the wetlands, however, C4 sedges may make up to 60% of the vegetation cover at 550 m. At higher altitudes the coverage of C4 sedges declines [Kotze & O'Connor 2000].

Cyperaceae pollen concentration (Figure 3) and percentages correlate with ln(Fe/Ca) and with $\delta^{13}C_{wax}$ (Table 1, Figure 2). The ratios of terrestrial iron over marine calcium can be interpreted as a measure for terrestrial input, which in this part of the ocean is mainly fluvial. Correlation between increased fluvial discharge and increased C4 vegetation as well as increased Cyperaceae pollen has been reported from sediments off the Zambezi [Schefuß et al. 2011, Dupont & Kuhlmann 2017]. Moreover, a fingerprint of C4 sedges was found in Lake Tanganyika [Ivory & Russel 2016]. As a consequence, material (leaf waxes and pollen) from the riverine vegetation is probably better represented than that from dry and upland vegetation. These results corroborate the reinterpretation of the $\delta^{13}C_{wax}$ record, in which the increased representation of C4 plants (*n*-alkanes enriched in $^{13}C$) is instead attributed to stronger transport of material from the upper Limpopo catchment and the extension of swamps containing C4 sedges under more humid conditions [Caley et al., 2018]. Previously Castañeda et al. [2016] had interpreted increased C4 inputs as reflecting increased aridity.

Relatively low values of Cyperaceae pollen and Fe/Ca ratios are found for most interglacials of the Brunhes Chron (Figures 2 and 3), which could be interpreted as an effect of sea-level high-stands. However, Caley et al. [2018] demonstrated that the fluvial discharge is not related to sea-level changes. From the bathymetry of Delagoa Bight, strong influence of sea-level is also not expected because the shelf is not broad and the locality of Core MD96-2048 is relatively remote on the Limpopo cone in the center of the clockwise flowing Delagoa Bight Lee Eddy. The eddy transports terrestrial material northeastwards before it is taken southwards (Figure 1) [Martin 1981] and likely has not changed direction during glacial times. Thus, fluvial discharge was probably low during interglacials (among other periods), which might be the combined result of more evapotranspiration and less precipitation. Despite drier conditions, the representation of woodland and dry forest is relatively high during the interglacial periods (Figure 3, see also next section).

## Endmembers representing vegetation on land

Palynological results have been published for the past 350 ka [Dupont et al. 2011] providing a detailed vegetation record for the past three climate cycles. Pollen and spore assemblages could be characterized initially by three endmembers via endmember modelling (EM1, EM2, EM3). The assemblage of EM1 was dominated by *Podocarpus* (yellow wood) pollen percentages being more abundant during the non-interglacial parts of MIS 5, 7, and 9. EM2 was characterized by pollen percentages of Cyperaceae (sedges), Ericaceae (heather) and other plants of open vegetation and abundant during full glacial stages. EM3 constituted of pollen from woodland, forest, and coastal vegetation and was interpreted to represent a mix of several vegetation complexes.

We repeated the endmember modelling for the extended record covering the entire Brunhes Chron
and the two early Pleistocene windows. The analysis of the extended dataset gave compatible results
with the previous analysis (Dupont et al. 2011). The main difference is that the longer sequence
allowed to distinguish two assemblages of interglacial vegetation. In terms of analysis, the
cumulative increase of explanatory power lessened after four (instead of three) endmembers and a
model with four endmembers was chosen. We used the scores of the different pollen taxa on the
endmember assemblages for our interpretation of the endmembers (list of taxa and scores in
Supplementary Tables 1 and 2). This interpretation is summarized in Table 2. To distinguish between
the previous and current analysis (which show strong similarities), we have given new names to the
endmember assemblages reflecting our interpretation: E-heathland, E-Mountain-Forest, E-
Shrubland, E-Woodland. A selection of pollen percentage curves are plotted together with each
endmember's fractional abundance in Supplementary Figures 1-4.

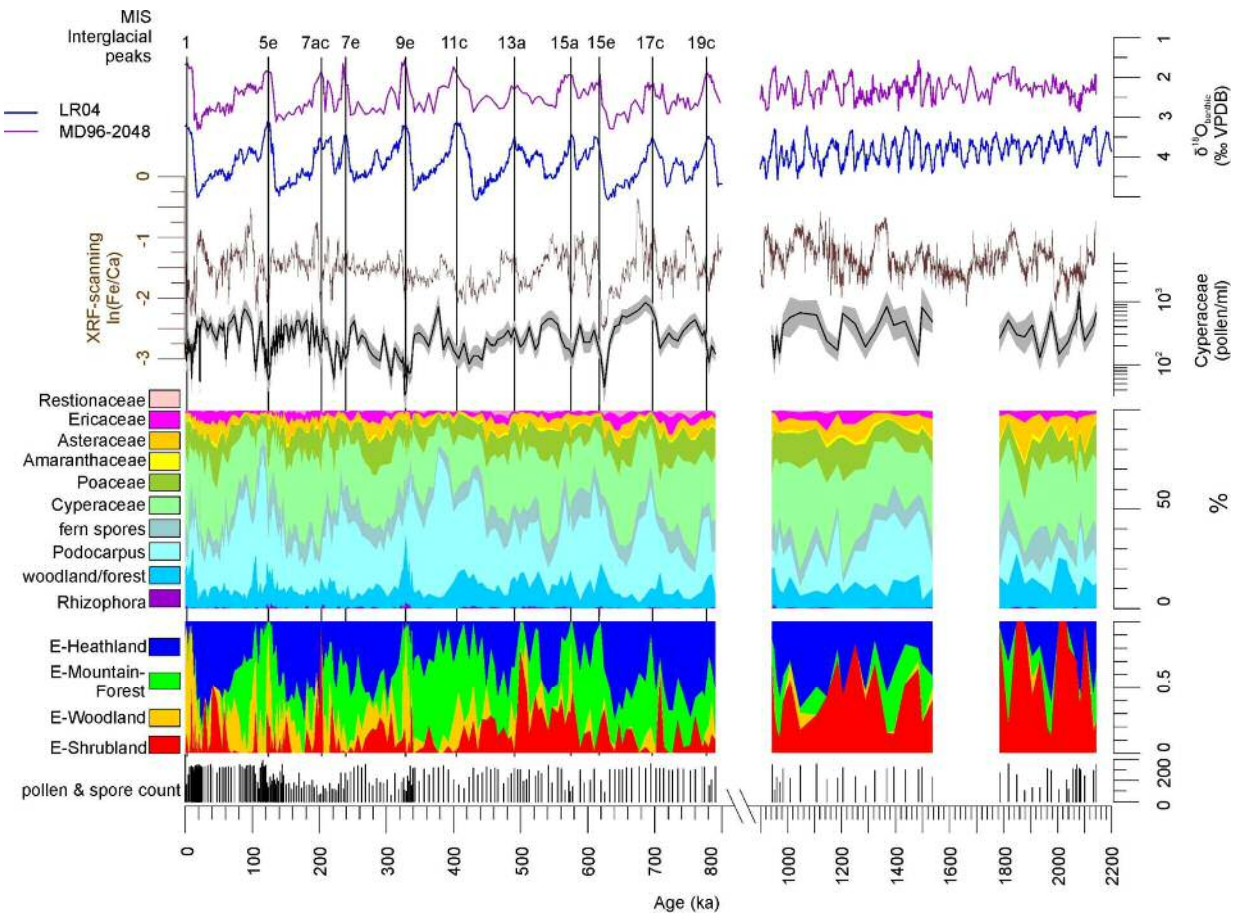

Figure 3. Summary of results of MD96-2048. Bottom to top: Pollen and spore count used to calculate percentages; Fractional abundance of endmembers E-Shrubland, E-Woodland. E-Mountain-Forest, and E-Heathland; Pollen summary diagram (woodland and forest taxa are listed in Supplementary Table 1; Cyperaceae pollen concentration per ml (shading denoted 95% confidence intervals after Maher [1981]); ln(Fe/Ca); global stack of stable oxygen isotopes of benthic foraminifers, LR04 [Liesiecki & Raymo 2005]; $\delta^{18}O_{benthic}$ of Core MD96-2048 [Caley et al. 2018]; Interglacial peaks after PAGES [2016]. VPDB: Vienna Pee Dee Belemnite.

**E-Heathland.** Of the four endmember assemblages (Figure 3), one endmember had a counterpart in
EM2 [Dupont et al. 2011] of the previous analysis. Not only composition but also the fractional
abundances, which were high during glacial stages, are very much alike. We name this endmember
'E-Heathland', which is dominated by Cyperaceae (sedges) pollen percentages followed by Ericaceae
(heather) pollen and hornwort (Anthocerotaceae) spores (Table 2). Also *Lycopodium* (clubmoss)
spore, Restionaceae and *Stoebe*-type pollen percentages score highest on this endmember. The E-
Heathland assemblage represents a Fynbos-like open vegetation growing during full glacials. Other
pollen records from SE Africa also indicate an open ericaceous vegetation with sedges and
Restionaceae during glacial times [Scott 1999, Dupont & Kuhlmann 2017]. The record of MD96-2048
testifies that this type of open glacial vegetation regularly occurred since at least two million years.
**E-Mountain-Forest.** Like the endmember EM1 [Dupont et al. 2011] of the previous analysis, one
endmember is dominated by *Podocarpus* (yellow wood) pollen percentages (Table 2). The
assemblage is enriched by pollen of *Celtis* (hackberries) and *Olea* (olive trees) accompanied by
undifferentiated fern spores. The interpretation as an assemblage representing mountain forest is
rather straightforward and we name the assemblage 'E-Mountain-Forest'. The fractional abundance
of the E-Mountain-Forest is also high in glacials of the Brunhes Chron but not during the extreme
glacial stages, when temperatures and $pCO_2$ are particularly low (Figure 3). It is low in the early -
Pleistocene parts of the record.
**E-Shrubland.** The remaining endmember assemblages have no direct counterpart in the previous
analysis, although summed together the pattern of fractional abundance is similar to that of EM3
[Dupont et al. 2011]. One endmember groups together 44 pollen taxa, mostly from coastal and dune
vegetation, which we name 'E-Shrubland'. It includes pollen of Asteraceae and Poaceae (grasses).
The latter are not very specific as grass pollen values score almost as high on other endmember
assemblages (E-Heathland and E-Woodland). Several taxa scoring on this endmember are known
from coastal or halophytic settings such as *Gazania*-type, Amaranthaceae, *Tribulus*, Acanthaceae and
*Euphorbia*-type. Arboreal taxa in this assemblage are *Dombeya*, *Acacia*, Meliaceae/Sapotaceae (Table
2). The most typical taxa are the *Buxus* species. We distinguished three types of *Buxus* pollen: *B.*
*macowani* type, *B. hildebrandtii* type and *B.* cf. *madagascarica* [Köhler & Brückner 1982, 1989]. *B.*
*madagascarica* grows on Madagascar and its pollen is only found sporadically, while the other two
species inhabit bushland and forest on coastal dunes of the East African main-land. *B. hildebrandtii*
nowadays is found in Somalia and Ethiopia and *B. macowani* is native in South Africa. The record of
M96-2048 indicates that these *Buxus* species were more common in the early Pleistocene than
during the Brunhes Chron (Figure 3).

Table 2. Interpretation of the endmembers

| Endmember | Main pollen taxa |
|---|---|
| E-Heathland | *Podocarpus*, *Celtis*, *Olea* |
| E-Mountain-Forest | Cyperaceae, Ericaceae, *Phaeoceros*, Restionaceae, *Stoebe* type, *Anthoceros*, *Typha Lycopodium*, Restionaceae |
| E-Shrubland | Poaceae, Asteroideae, *Buxus*, Amaranthaceae, *Euphorbia*, Meliaceae-Sapotaceae, *Acacia*, *Riccia* type, *Tribulus*, Acanthaceae pp, Asteraceae Vernoniae, *Hypoestes-Dicliptera* type, *Gazania* type, *Dombeya* |
| E-Woodland | *Alchornea*, *Spirostachys africana*, *Pteridium* type, Polypodiaceae, *Myrsine africana*, *Cassia* type, Rhizophoraceae, *Aizoaceae*, Combretaceae pp, *Manilkara*, *Burkea africana*, *Brachystegia*, *Dodonaea viscosa*, *Pseudolachnostylis*, *Hymenocardia*, *Aloe*, Rhamnaceae pp, *Protea*, *Parinari* |


**E-Woodland.** The last endmember, which we name 'E-Woodland', groups together 39 pollen taxa
from forest and woodland species with maximum values of less than 5 or 2% of the total of pollen
and spores. To this assemblage belong *Pseudolachnostylis, Dodonaea viscosa* and *Manilkara*, which
are woodland trees. *Protea* (sugarbush) and *Myrsine africana* (Cape myrtle) grow more upland and
*Alchornea* is a pioneer forest tree often growing along rivers. Others include wide-range woodland
taxa such as Combretaceae species. The occurrence of pollen of *Brachystegia* (miombo tree), *Burkea*
*africana*, *Spirostachys africana* and *Hymenocardia* in this assemblage is indicative of Miombo dry

forest and woodland. The assemblage additionally includes Rhizophoraceae pollen from the coastal
mangrove forest (Table 2). The fractional abundance of the E-Woodland assemblage is low during the
early Pleistocene, increased during the interglacials prior to the MBE and had maximum values
during Interglacials 9e, 5e, and 1 (Figures 3, 4). These interglacials also exhibited maximum
percentages of arboreal pollen excluding *Podocarpus*.

In summary, the endmember analysis indicates a very stable open ericaceous vegetation with
partially wet elements such as sedges and Restionaceae characterizing the landscape of full glacials
(when global temperatures and $p$CO$_2$ were lowest). During the less extreme parts of the glacials,
mountain *Podocarpus* forest was extensive as in most mountains of Africa [Dupont 2011, Ivory et al.
2018]. On the other hand, interglacials were characterized by coastal shrubs. In the course of the
Brunhes, the woody component, which was relatively weak before the MBE, became more and more
important reflecting the same long-term trend found in the leaf wax records [Castañeda et al. 2016].
It is likely that the Miombo dry forest and woodland migrated into the region in the successive
interglacials of the Brunhes Chron. Particularly during Interglacials 9e and 1 the area might have been
more forested than during the older interglacials of the Brunhes Chron.

## Long-term trends in vegetation and climate of East Africa

The region of the Limpopo River becoming more and more wooded in the course of successive
interglacials [Castañeda et al. 2016] somewhat paralleled the conditions around Lake Malawi
[Johnson et al. 2016]. However, around Lake Malawi, forested phases of either mountain forest,
seasonal forest, or Miombo woodland alternating with savannahs occurred during both glacial and
interglacial stages [Ivory et al. 2018]. Also in contrast to the Lake Malawi record, the MD96-2048
Poaceae pollen percentages fluctuated little and remained relatively low (less than 20%) indicating
that savannahs were of less importance in the Limpopo catchment area and the coastal region of
southern Mozambique.

The trend to increased woodland in SE Africa after the MBE, noted at both Lake Malawi and in the
Limpopo River catchment [Johnson et al. 2016, Caley et al. 2018, this study] contrasts with the trend
around Lake Magadi at the equator. At Lake Magadi a trend to less forest around marks the Mid-
Brunhes transition [Owen et al. 2018]. Antiphase behavior of SE African climate with that of West
and East Africa emphases the importance of the average position of the tropical rainbelt shifting
southwards during globally cold periods as has been inferred from Holocene to Last Glacial records of
Lake Malawi [Johnson et al. 2002, Scholz et al. 2011]. Our results confirm this relationship existed
over the entire Brunhes Chron.

The Lake Malawi pollen record as well as that of the equatorial Lake Magadi in Kenya [Owen et al.
2018] do not show much of a glacial-interglacial rhythm and are dominated by the precession
variability in tropical rainfall  [cf. Clement et al. 2004]. Obviously, in the tropical climate of the
Southern Hemisphere north of ~15°S the hydrological regime had more effect on the vegetation than
changes in temperature, while further south the impact of glacial-interglacial variability on the
vegetation increased.

## Effects of atmospheric $p$CO$_2$

While the hydroclimate of the region shows precession variability [Caley et al. 2018], the vegetation
shows a glacial-interglacial rhythm (Supplementary Information) indicating that besides hydrology,
temperature and/or atmospheric CO$_2$ levels were important drivers of the vegetation development.
Combining the results of the pollen assemblages with stable carbon isotopes and elemental
information indicates that during interglacials the region of SE Africa (northern South Africa,

Zimbabwe, southern Mozambique) was less humid. This is in accordance with other paleoclimate
estimates for the region [see reviews by Simon et al. 2015, Singarayer & Burrough 2015].
The interglacial woodlands (represented by E-Woodland, Figure 4) would probably have grown under
warmer and drier conditions than the glacial mountain forest (represented by E-Mountain-Forest).
The increase in maximum $pCO_2$ levels during the post-MBE interglacials might have favored tree
growth as higher $pCO_2$ levels would have allowed decreased stomatal conductivity and thus relieved
drought stress [Jolly & Haxeltine 1997]. Woodlands would have expanded at the cost of mountain
forest during 11c, 9e, 5e and 1, and to a lesser extend during 7e and 7c, when temperatures and
$pCO_2$ were high (Figure 4). It might be only after interglacial $pCO_2$ levels rose over ~270 ppmv that
Miombo woodland could fully establish in the area during the warm and relatively dry post-MBE
interglacials.
The glacial stages showed the expansion of either mountain forest or heathland. The record indicates
extension of mountain forests in SE Africa during those parts of the glacial stages with low
temperatures and atmospheric $pCO_2$ exceeding ~220 ppmv (Figure 5).  If low temperatures were the
only driver of the extension of mountain forests, further spread into the lowlands during the coldest
glacial phases should be expected. Instead, when $pCO_2$ dropped below ~220 ppmv during those
colder glacial periods, mountain forest declined, in particular during MIS 18, 16, 14, 8, 6, and 2. A

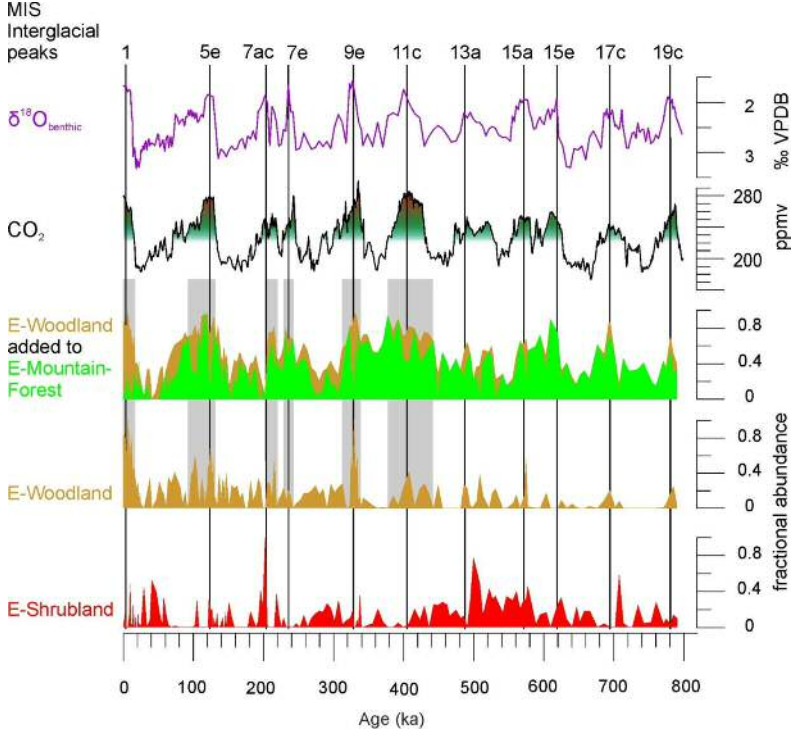

Figure 4: Comparing pollen assembages E-Mountain-Forest, E-Woodland and E-Shrubland with atmospheric $CO_2$ [Bereiter et al. 2015, PAGES 2016]. On top Interglacial peaks of the past 800 ka [PAGES 2016] and stable oxygen isotopes of benthic foraminifera ($\delta^{18}O_{benthic}$) of MD96-2048 [Caley et al. 2011]. $CO_2$-levels of 220 and 270 ppmv are indicated with green-red shading. Grey shading highlight periods with maximum atmospheric $CO_2$ and maximum values of the sum of E-Woodland and E-Mountain-Forest. VPDB: Vienna Pee Dee Belemnite.

picture emerges of cool glacial stages in SE Africa in which tree cover broke down when atmospheric
$pCO_2$ became too low. Additionally, mountain forests were important during the Interglacials 19c,
17c, 15e, 15a, 13a, and 7e, in which $pCO_2$ and Antarctic temperatures were subdued.

With an inverse modelling technique, Wu et al. [2007] estimated the climate inputs for the
vegetation model BIOME4 using as information the biome scores of pollen records from equatorial
East African Mountains. Wu et al. found that lowering of the tree line under glacial conditions (1-3°C
lower temperatures, less precipitation, 200 ppmv $p\text{CO}_2$) depended hardly on temperature but
primarily on increased aridity and somewhat on lower $p\text{CO}_2$, whereby lower $p\text{CO}_2$ amplified the
effects of water limitation. However, Izumi & Lézine [2016] found contrasting results using pollen
records of mountain sites on both sides of the Congo basin. At any rate, the lack of trees in the
Southeast African Mountains during glacial extremes is unlikely the result of drought, because our
record indicates that climate conditions in SE Africa were less dry during glacials than during
interglacials (the post-MBE interglacials in particular). Instead, C4 sedges being an important
constituent of the ericaceous fynbos-like vegetation increased during glacials when atmospheric
$p\text{CO}_2$ and temperatures were low (Figure 5). However, low temperatures are not particularly
favorable for C4 sedges as indicated by the altitudinal distribution of C4 sedges in modern wetlands
of KwaZulu Natal [Kotze & O'Connor 2000]. We presume, therefore, that the extension of C4 sedges
during the more humid phases of the glacials is the result of low atmospheric $\text{CO}_2$ concentrations
rather than of low temperatures.

Pollen records of ericaceous vegetation suggest an extensive open vegetation existing in the East
African Mountains [e.g. Coetzee 1967, Bonnefille & Riollet 1988, Marchant et al. 1997, Debusk 1998,
Bonnefille & Chalié 2000] and in SE Africa and Madagascar [e.g. Botha et al. 1992, Scott 1999, Gasse
and Van Campo 2001, Scott & Tackeray 1987] during the last glacial. In our study, ericaceous fynbos-
like vegetation (E-Heathland) was found for those parts of the glacials having lower (less than ~220
ppmv) atmospheric $p\text{CO}_2$ (Figure 5). Exceptions were found for MIS 12 and 14 when the difference of
$p\text{CO}_2$ with that of the preceding stage was small [Bereiter et al. 2015]. Dupont et al. [2011] argued
that increase of C4 vegetation as the result of low $p\text{CO}_2$ was unlikely because no extension of grasses
was recorded. However, this argument is flawed if sedges dominantly constituted the C4 vegetation
in the area. We also note that in many parts of South Africa, no substantial increase of C4 grasses
occurred but that many sites suggest an expansion of C3 grasses during the Last Glacial Maximum
[Scott 2002].

As climate was wetter during most of the glacials in this part of the world, the question arises about
the climatic implication of the ericaceous fynbos-like vegetation (represented by E-Heathland, Figure
5) extending during full glacials over the mountains of South Africa - and correlating with the SST
record (see also the correlation between SST and EM2 in Dupont et al. 2011). The correlation with
SST, however, is problematic. Singarayer & Burrough [2015] argued that the control of the Indian
Ocean SSTs on the precipitation of South Africa shifted from a positive correlation during the
interglacial to a negative correlation during the Last Glacial Maximum. They invoked the effects of
the exposure of the Sunda Shelf (Indonesia) and Sahul Shelf (Australia) on the Walker circulation
causing a wetter region over the western Indian Ocean but also weaker easterly winds to transport
moisture inland. To question the link between SST and precipitation in SE Africa even further, Caley
et al. [2018] found that the precession signature in the river discharge proxy [ln(Fe/Ca), see also
Supplementary Information] was absent in the SST record from the same core. SE Africa would have
been more humid during glacials when the temperature difference between land and sea increased.

The increase in C4 vegetation during relative cool and humid climate would be in conflict with the
idea that C4 plants are more competitive in hot and dry climates [Ehleringer et al. 1997, Sage 2004].
However, this idea is mainly based on the ecology of grasses and the development of savannahs,
while the C4 vegetation expansion in SE Africa during cool and humid phases seems to be driven by
sedges. A survey of the distribution of C4 sedges in South Africa revealed that those Cyperaceae do
not have the same temperature constraints as C4 grass species [Stock et al. 2004]. More important,
South African C4 sedges appear to have evolved under wetland conditions rather than under aridity.
C4 *Cyperus* species even occur in the wettest parts of lower altitude wetlands in KwaZulu-Natal
[Kotze & O'Conner 2000].

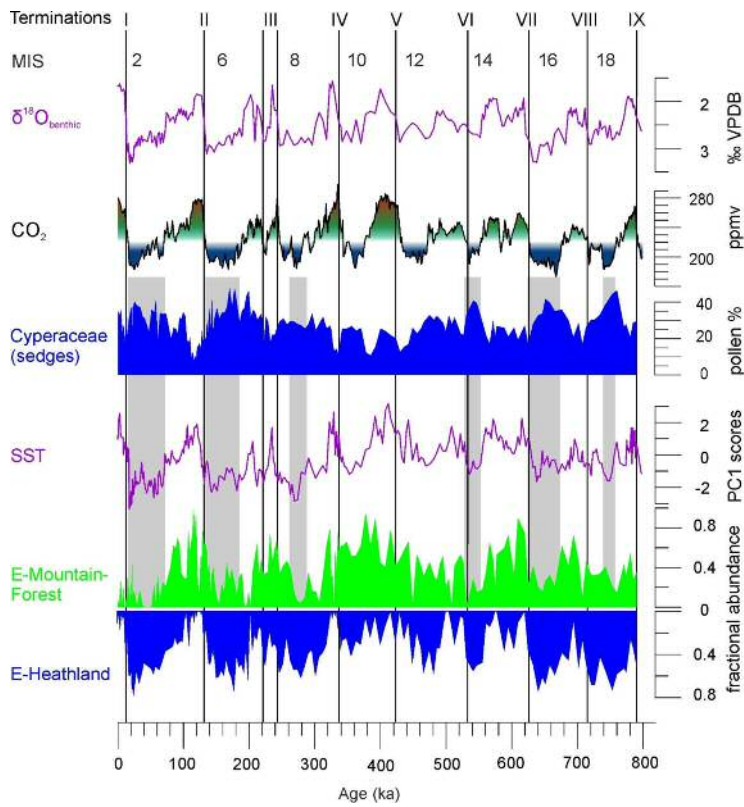

Figure 5: Comparing Cyperaceae pollen percentages and fractional abundances of the glacial pollen assembages E-Mountain-Forest and E-Heathland with atmospheric $CO_2$ [Bereiter et al. 2015, PAGES 2016] and sea surface temperatures of the southeastern Indian Ocean (SST PC1 scores of MD96-20468) [Caley et al. 2018]. On top Terminations of the past 9 glacacions, glacial marine isotope stages (MIS) and stable oxygen isotopes of benthic foraminifera ($\delta^{18}O_{benthic}$) of MD96-2048 [Caley et al. 2011]. $CO_2$-levels of 220 and 270 ppmv are indicated with blue-green-red shading. Grey shading highlight periods with minimum atmospheric $CO_2$, minimum values of E-Mountain-Forest, and maximum values of E-Heathland and Cyperaceae pollen. VPDB: Vienna Pee Dee Belemnite.

# 10  Conclusions

Palynology in combination with sediment chemistry and carbon isotope analysis of leaf waxes carried
out on the marine sediments of MD96-2048 retrieved from the Limpopo River cone in the Delagoa
Bight (SE Africa) allowed a detailed reconstruction of the biome developments over the Brunhes
Chron and comparison with earlier Pleistocene vegetation of SE Africa.
Using endmember modelling, we could distinguish four pollen assemblages: E-Heathland, E-
Mountain-Forest, E-Shrubland, E-Woodland. The open sedge-rich and ericaceous vegetation

represented by E-Heathland ocurred during those parts of the glacials with lower temperatures and atmospheric $p$CO$_2$. *Podocarpus*-rich mountain forest represented by E-Mountain-Forest extended during the less extrem parts of the glacials. E-Shrubland represents a shrublike vegetation with coastal elements and *Buxus* species and mainly occurred during the earlier Pleistocene (before 1 Ma). E-Woodland represents interglacial woodlands, Miombo woodland in particular, becoming more and more important in the succesive interglacial stages of the Brunhes Chron and dominated the post-MBE interglacials.

Our results indicate the influence of atmospheric $p$CO$_2$ fluctuations on the shaping of the biomes in SE Africa during the Brunhes Chron. We argue that (1) the precessional rhythms of river discharge compared to the interglacial-glacial biome variability indicates that hydroclimate cannot be the only driver of vegetation change. The other options of forcing mechanisms on interglacial-glacial time-scales are temperature and $p$CO$_2$. (2) Because of the correlation between Cyperaceae pollen percentages and $\delta^{13}C_{wax}$ and the lack of correlation between Poaceae percentages and $\delta^{13}C_{wax}$ in combination with the relatively low grass pollen percentages, we deduce that the C4 plant imprint mainly derives from the sedges. (3) The expansion of C4 sedges during the colder periods of the glacials is unlikely to result from lower temperatures. Thus, during the colder phases of the glacials, low atmospheric $p$CO$_2$ might have favored the expansion of C4 sedges. (4) The confinement of mountain forest to the glacial periods with moderate temperatures and moderate $p$CO$_2$, and the lack of extension into the lowlands of mountain forest during the colder periods, suggests that low $p$CO$_2$ became restrictive to the forest. Moutain forests could thrive during glacials as long as $p$CO$_2$ levels exceeded ~220 ppmv. (5) Based on the elemental composition as a proxy for river discharge [ln(Fe/Ca)], we recognise the post-MBE interglacials as the drier intervals of the sequence. Nevertheless woodland extended during those periods, which we attribute to increased temperatures and $p$CO$_2$. Atmospheric $p$CO$_2$ levels over 250 ppmv might have been a prerequisite for the establishment of the Miombo woodlands in SE Africa, which extended during the post MBE interglacials.

The vegetation record of the Limpopo catchment area shows a greater impact of glacial-interglacial variability, mainly driven by CO$_2$ fuctuations, and less influence of hydroclimate compared to the more equatorial records of Lake Malawi and Lake Magadi. The long-term trend of increased woodiness in the course of the Brunhes Chron parralleled that of Lake Malawi but constrasted Lake Magadi suggesting a long-term southward shift in the average position of the tropical rainbelt.

## Acknowledgements

This project was funded through DFG -Research Center / Cluster of Excellence „The Ocean in the Earth System". T.C. is supported by CNRS-INSU. Funding from LEFE IMAGO CNRS INSU project SeaSalt is acknowledged. David Heslop is thanked for providing the Endmember model into a Matlab application.

## Data availability

Pollen counts are available at https://doi.pangaea.de/10.1594/PANGAEA.897922.

Previously published data can be retrieved at https://doi.pangaea.de/10.1594/PANGAEA.895364;
https://doi.pangaea.de/10.1594/PANGAEA.895361;
https://doi.pangaea.de/10.1594/PANGAEA.895362;
https://doi.pangaea.de/10.1594/PANGAEA.863919;
https://doi.pangaea.de/10.1594/PANGAEA.895357.

# Author contributions

LMD carried out the palynological analysis, concepted and wrote the manuscript, TC carried out the sedimentology and stratigraphy and contributed to the discussion, ISC conducted the stable isotope analysis on plant waxes and contributed to the discussion.

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

# 1 Supplement

REDFIT frequency analysis
We conducted a frequency analysis on the data of the ln(Fe/Ca) ratios, the E-Heathland fractional
abundance scores, and the Cyperaceae pollen concentration covering the Brunhes Chron using the
algorithm of REDFIT [Schulz & Mudelsee 2002] from the statistical package PAST version 3.14 (1999-
2006) [Hammer et al. 2001]. The E-Heathland and Cyperaceae curves each consisted of 181 data
points between 0 and 790 ka. REDFIT was run with 2 times oversampling, a Blackman-Harris window,
and 2 overlapping averaging segments resulting in a bandwidth of 0.004291; false alarm level was
99.17. The ln(Fe/Ca) curve contained 2307 data points between 1 and 790 ka. REDFIT was run with 2
times oversampling, a Blackman-Harris window, and 3 overlapping averaging segments resulting in a
bandwidth of 0.005726; false alarm level was 99.91. The figure shows the power of ln(Fe/Ca) ratios
(left), the power of the E-Heathland values (middle), and the power of the Cyperaceae pollen
concentration (right) against frequency running from 0 - 0.08 cycles per ka. Denoted are the
bandwidth for each spectrum and a parametric approximation of the level above the null hypothesis
of a red noise model using $X^2$-test at 90% (dashed lines). Grey bars indicate the orbital periodicities of
100, 41, 23, and 19 ka). Note the maximum in spectral density at 23 ka (precession) in the power
spectrum of ln(Fe/Ca) and the lack of spectral density at the precession bands (23 and 19 ka) in the
power spectrum of the E-Heathland values. The Cyperaceae pollen concentration, which is both
influenced by the expansion of Cyperaceae (sedges) and by the transport of pollen by river discharge,
shows significant power at both the 100 and 19 ka.

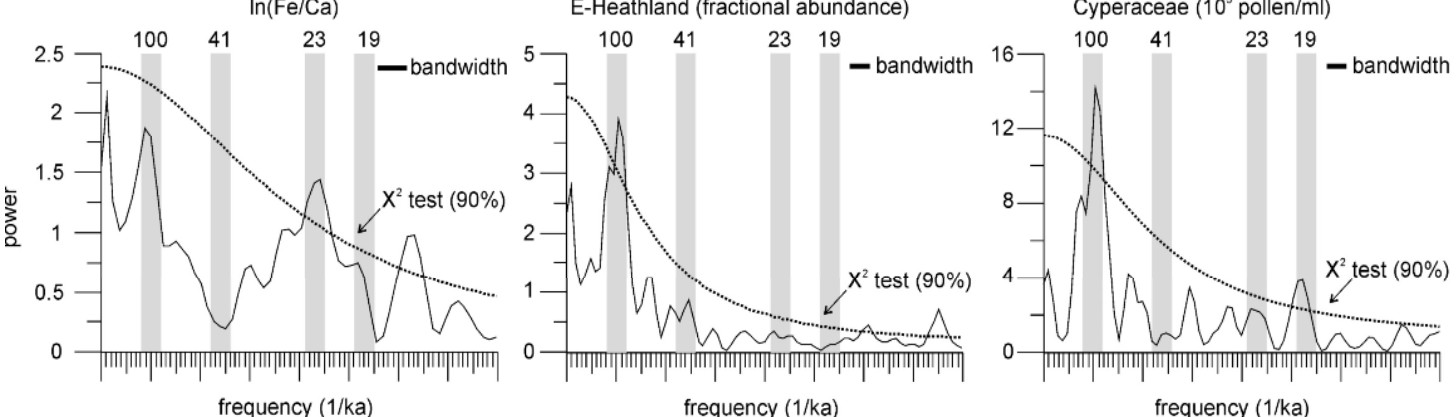

# 22 Supplementary References

Hammer, Ø., Harper, D.A.T. & Ryan, P.D., 2001. PAST: Paleontological Statistics Software Package for Education and Data
Analysis. Palaeontologia Electronica, 4(1): 1-9.
Schulz, M. & Mudelsee, M., 2002. REDFIT: estimating red-noise spectra directly from unevenly spaced paleoclimatic time
series. Computers & Geosciences, 28: 421-426.
Excel-Tables
Supplementary Table 1: Family, Pollen taxon, group, growth form, endmember assemblage of
maximum score.
Supplementay Table 2: Family, Pollen taxon, group, growth form, score per endmember (E-
Mountain-Forest, E-Heathland, E-Woodland, E-Shrubland), significance of taxon scores ($r^2$) for two
endmember analyses [this study and Dupont et al. 2011].
Supplementary Figures 1-4

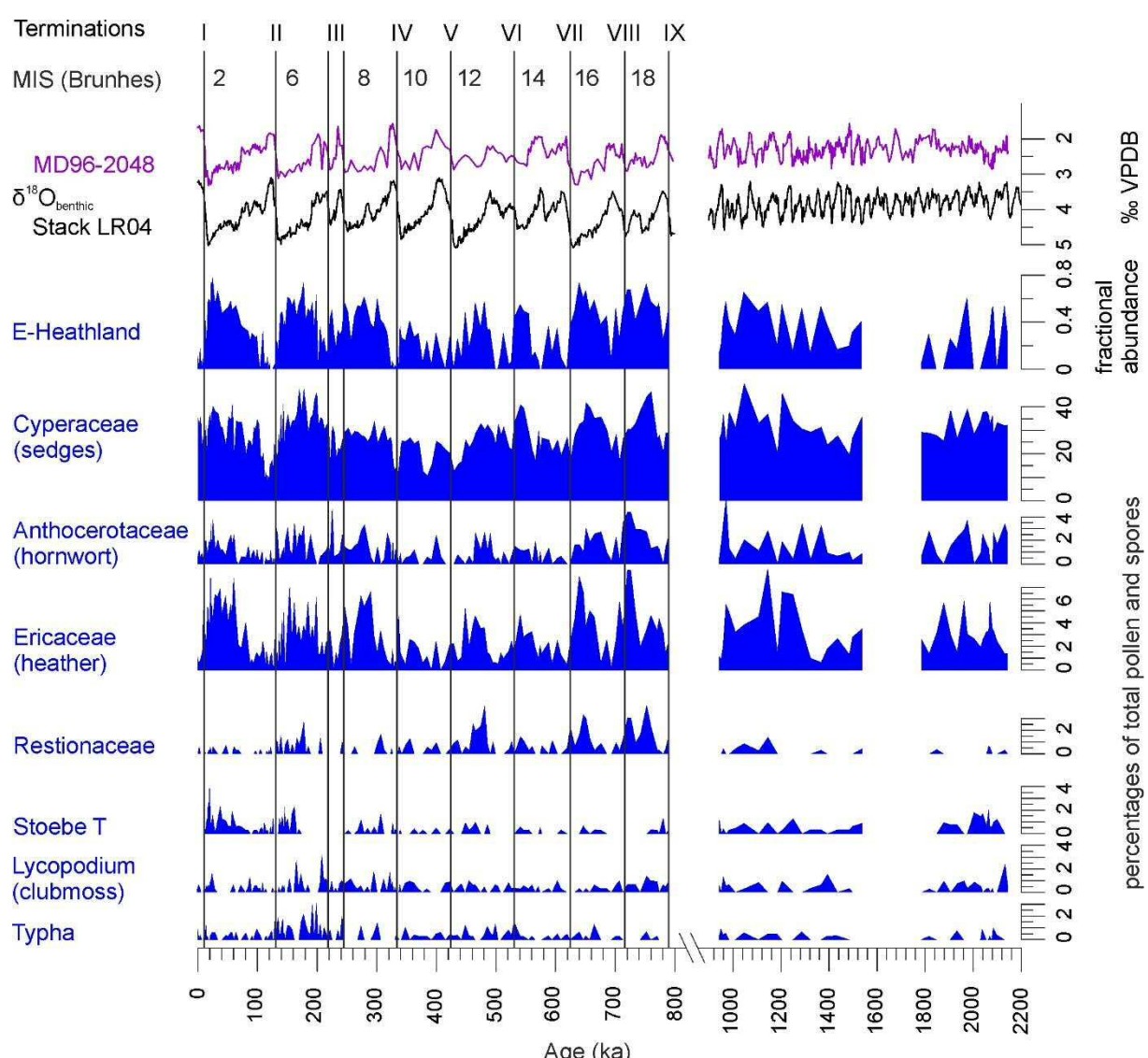

Supplementary Figure 1. Endmember assemblage scores and selected pollen taxa for E-Heathland (filled curves) against age
in ka. On top Terminations and even-numbered marine isotope stages (MIS) of the Brunhes Chron are indicated. Stable
oxygen isotopes of benthic foraminifers of Core MD96-2048 (violet line) and of global stack LR04 (black line, Lisiecki &
Raymo 2005].

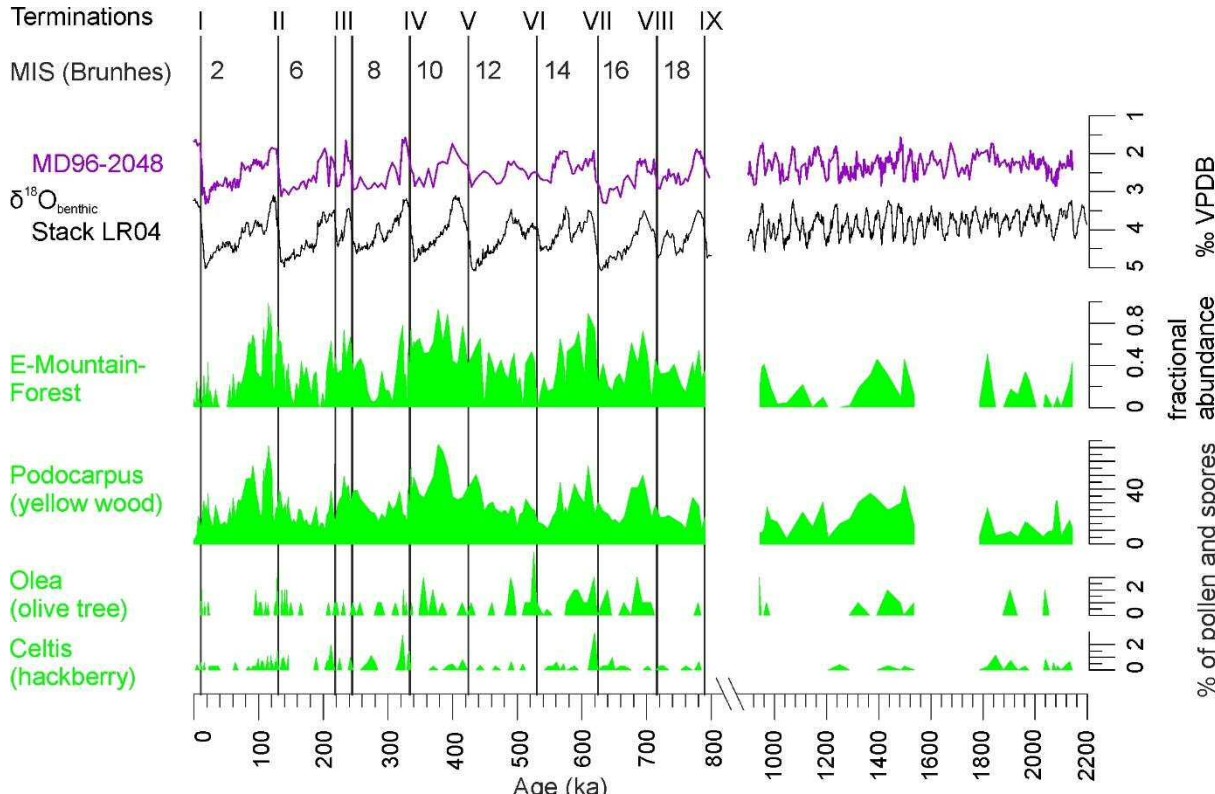

Supplementary Figure 2. Endmember assemblage scores and selected pollen taxa for E-Mountain-Forest (filled curves) against age in ka. On top Terminations and even-numbered marine isotope stages (MIS) of the Brunhes Chron are indicated. Stable oxygen isotopes of benthic foraminifers of Core MD96-2048 (violet line) and of global stack LR04 (black line, Lisiecki & Raymo 2005].

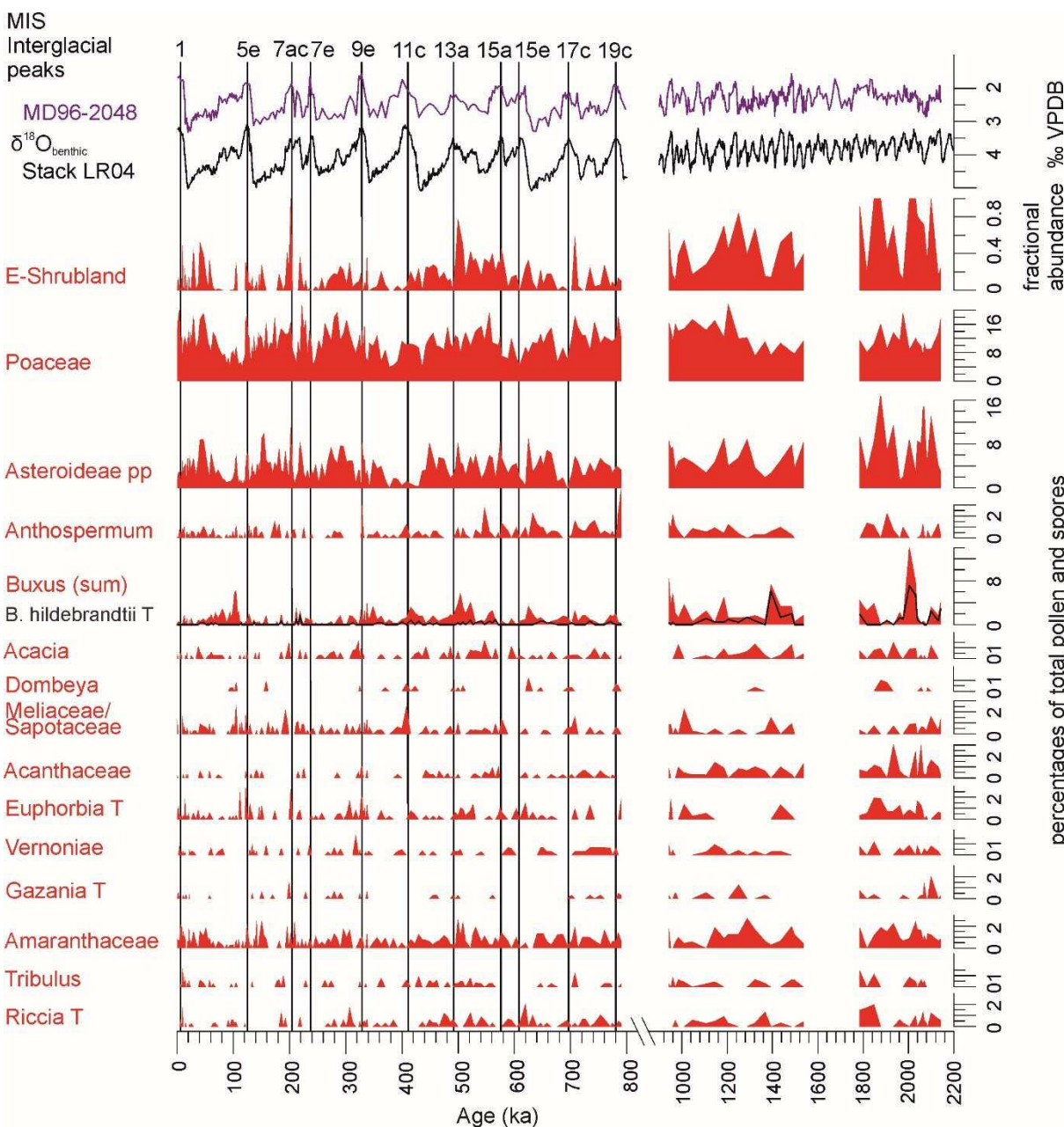

Supplementary Figure 3. Endmember assemblage scores and selected pollen taxa for E-Shrubland (filled curves) against age in ka. On top Terminations and even-numbered marine isotope stages (MIS) of the Brunhes Chron are indicated. Stable oxygen isotopes of benthic foraminifers of Core MD96-2048 (violet line) and of global stack LR04 (black line, Lisiecki & Raymo 2005].

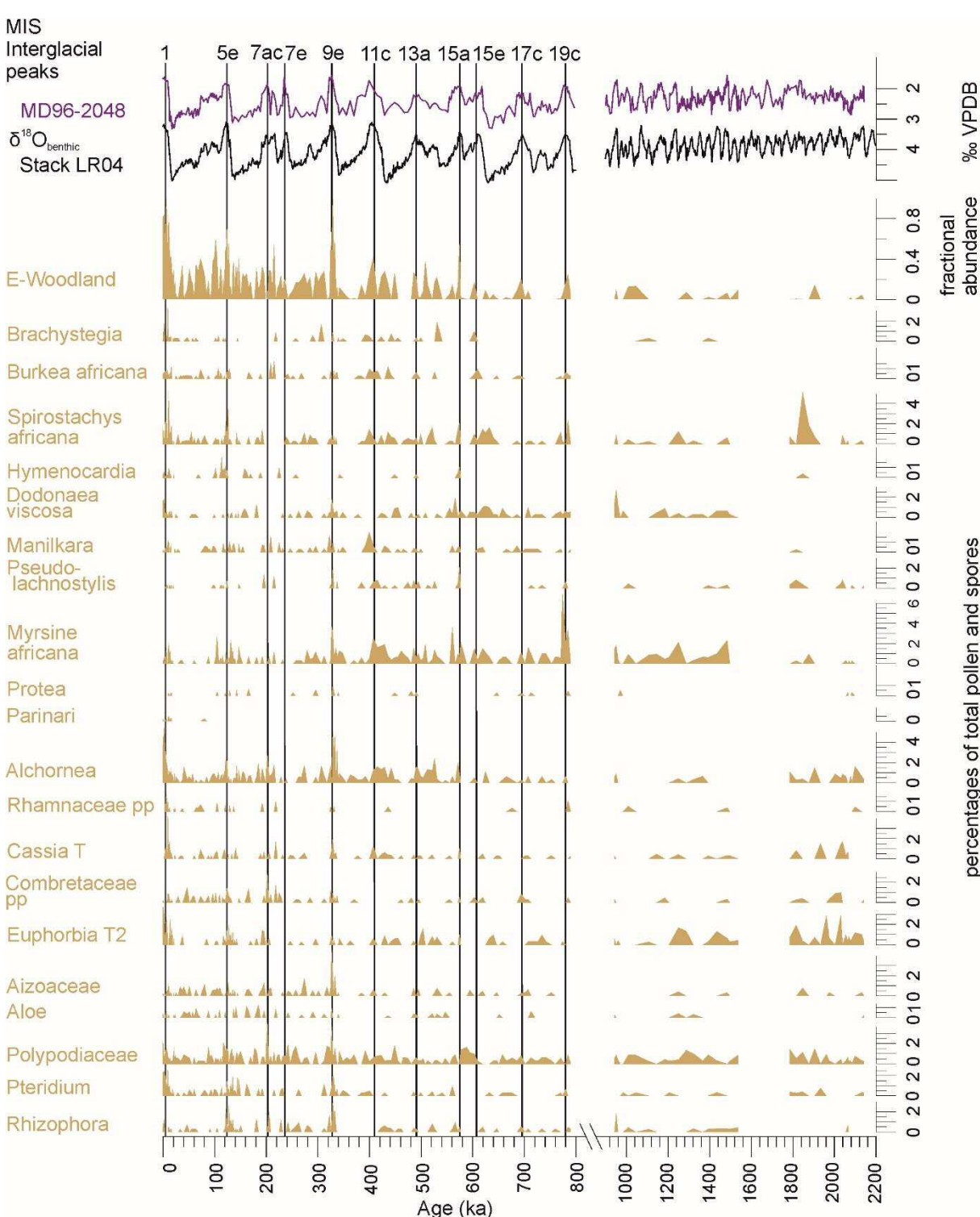

Supplementary Figure 4. Endmember assemblage scores and selected pollen taxa for E-Woodland (filled curves) against age
in ka. On top Terminations and even-numbered marine isotope stages (MIS) of the Brunhes Chron are indicated. Stable
oxygen isotopes of benthic foraminifers of Core MD96-2048 (violet line) and of global stack LR04 (black line, Lisiecki &
Raymo 2005].