# Peer review of "Effects of atmospheric $CO_2$ variability of the past 800 ka on the biomes of Southeast Africa"

_Climate of the Past, 2019_

## Referee Comment (RC1) · Anonymous Referee #1 · 21 Mar 2019

The paper reports on a prominent and unique pollen record from an Indian Ocean core and provides interesting comparisons with relevant other records derived from different sources including proxies from the same core and others like the Vostok Ice core. The paper is in connection with the influence atmospheric CO2 concentrations on eastern Southern African vegetation types and suggests that under the higher concentrations of pCO2 woody plant growth thrives. I can't say I am entirely convinced by the cause and effect arguments that are proposed between pCO2 and other factors like hyrdroclimate and temperature as is acknowledged by the authors, and therefore would suggest a more cautious approach. Proxies of these conditions seem to behave in concert but it is not that clear which one is the leading factor and how much they influence each another. Although the role C4 sedges seem to be important, I feel the role of C4

grasses are underplayed (discussed further below). This is nevertheless an important study worthy of publication provided that attention is given to the aspects listed below. Abstract P1, line 16. If possible, please mention to what degree and on what basis the effects of the factors, hydroclimate, temperature and atmospheric $pCO_2$, can be disentangled. P1, lines 17, 18, 19. The statement could provide better insight if it can be more specific, e.g., do these different vegetation categories respond in the same way or differently to $pCO_2$? The word "depended" might have to be reconsidered in view of the above concerns. Please insert ($\sim$430 ka) after Mid-Brunhes Event.

Introduction P1, line 24. If subscript is used in $pCO_2$, why not in C4? P2, line 13. It is unclear what kind of event is meant here. P2, line 20. The phrase "to counter" could be seen as ambiguous or is it meant to be "is counter to"?

Material and Methods P4, line 29. Does "windows" mean sections? A question arises here why there are two older windows and not one continuous one. Is there a hiatus or another reason? P5 line 11. A definition of an endmember would be helpful here to keep the uninitiated reader informed. P5, line 30. There may be correlation between sedge pollen and leaf wax isotopes but it looks as if there is also correlation with Poaceae pollen (see also below in connection with P9, line 15). P6, lines 6,7. Why is this relevant? Won't one find C4 sedges near any of these African lakes? P6, lines 19,20. I can't see why this is remarkable. In my experience dry conditions result in less ground cover hence relatively more woody elements.

Endmembers representing vegetation on land P6, line 23. In connection with the three endmembers, the reader would by now have seen from Figure 3 that there are 4 endmembers. Therefore, I suggest saying "characterized initially". The word "so-called" should have been used when endmembers were first mentioned. A table in the text with the most prominent constituents of the endmembers will make it easier to understand the significance without having to go and look in the supplements. P7, line 2. Must be "endmember's"? P7, line 3. This "one endmember" is a little confusing. Say which one or say: one endmember had a counterpart in EM2. P7. Line 15. What kind

of extremes? P7, line 30. "wide range woodland taxa" like Combretaceae? I see this important taxon in not mentioned except in the supplements. P8, line 8. This may be seen as ambiguous if "developed" is taken as originated. Did miombo not start much earlier?

Effects of atmospheric $pCO_2$ P9, line 15. Surely some C4 grasses also thrive under relatively moist conditions if it is not too cold. Seasonality or growing season moisture is a factor which was not evaluated enough in the paper. Is it not possible that apart from C4 sedges, certain C4 Poaceae also played a role? On high land, frost and winter seasonality might be a factor ruling out C4 grasses in favour of C3 grasses or small shrubs, but as long as there are summer rains and subtropical Africa is warm enough during the glacial periods, which will probably be the case on the low-lying coastal platform, C4 grasses will be supported. It might be worthwhile to consider Vogel's work on the distribution of C3/C4 grasses in Southern Africa in this study. There are also some arguments in this connection in Scott (2002). Vogel, J.C., Fuls, A., Ellis, R.P., 1978. The geographical distribution of Kranz grasses in South Africa. S. Afr. J. Sci. 74, 209-215. Scott, L. 2002. Grassland development under glacial and interglacial conditions in Southern Africa: review of pollen, phytolith and isotope evidence. Palaeogeography, Palaeoclimatology, Palaeoecology 177(1-2): 47-57.

---

## Short Comment (SC1) · 25 Mar 2019

Dupont et al. Climates of the Past The manuscript by Dupont et al. uses pollen assemblages and other multi-indicator data from a marine sediment core in order to look at vegetation and climate change in southeastern Africa over multiple glacial-interglacial cycles. They extend a previously published dataset to 800k and also include analysis of older mid and early Pleistocene sections in order to disentangle the effects of temperature, precipitation, and pCO2 on vegetation. I find this paper to present a very well done and important dataset from a data-sparse region of the tropics. I am really happy to see the pollen analyses extended to older sections of the core and appreciate the data-rich multi-indicator approach. I also support the end-member analysis conducted by the authors, which seems to be able to be used to tease apart assemblages through time. I also really appreciate the comparison of the carbon isotope data with pollen data, which I find enhances the interpretations from both datasets. I do however have a few concerns about the paper, these are mostly minor. In particular, I find that the main conclusions about pCO2 and its impact on vegetation over time is a bit thinly supported. No further data analysis supports this argument and in fact I find that the section which describes the patterns attributed to CO2 is very brief given the prominence of this driver in the abstract of the paper. My most pressing suggestion for improvement of this paper then is to further develop this section of the paper and perhaps include a plot showing major vegetation types under differing CO2 thresholds discussed. The wiggle plots which represent the bulk of this argument in Figures 4 and 5 are not sufficient for really teasing apart the impact of CO2 or illustrating the authors' interpretation.

See specific and minor comments in the attached file.

Please also note the supplement to this comment:
https://www.clim-past-discuss.net/cp-2019-18/cp-2019-18-SC1-supplement.pdf

**Supplement:**

Dupont et al. Climates of the Past

The manuscript by Dupont et al. uses pollen assemblages and other multi-indicator data from a marine sediment core in order to look at vegetation and climate change in southeastern Africa over multiple glacial-interglacial cycles.  They extend a previously published dataset to 800k and also include analysis of older mid and early Pleistocene sections in order to disentangle the effects of temperature, precipitation, and $pCO_2$ on vegetation.

I find this paper to present a very well done and important dataset from a data-sparse region of the tropics.  I am really happy to see the pollen analyses extended to older sections of the core and appreciate the data-rich multi-indicator approach.  I also support the end-member analysis conducted by the authors, which seems to be able to be used to tease apart assemblages through time.  I also really appreciate the comparison of the carbon isotope data with pollen data, which I find enhances the interpretations from both datasets.

I do however have a few concerns about the paper, these are mostly minor. In particular, I find that the main conclusions about $pCO_2$ and its impact on vegetation over time is a bit thinly supported.  No further data analysis supports this argument and in fact I find that the section which describes the patterns attributed to $CO_2$ is very brief given the prominence of this driver in the abstract of the paper. My most pressing suggestion for improvement of this paper then is to further develop this section of the paper and perhaps include a plot showing major vegetation types under differing $CO_2$ thresholds discussed.  The wiggle plots which represent the bulk of this argument in Figures 4 and 5 are not sufficient for really teasing apart the impact of $CO_2$ or illustrating the authors' interpretation.

*Page 1*

-Please switch to sequential line numbering for the whole paper, rather than per page, if that fits with journal formatting.

Line 26: I think this isn't quite accurate, because C4 vegetation is by its very nature arid adapted in that C4 metabolism is an adaptation to increase water use efficiency.  I think instead you should change this sentence to imply that the risk misinterpreting the bioclimatic controls on the expansion of arid adapted vegetation.
-Which studies?  Also this phrasing is awkward, I think "last Glacial-interglacial transition" or Pleistocene-Holocene transition might be better

Line 27: "Glacial" glacials?

*Page 2*

Line 4: "cycle" should be cycles

Line 11: Yin and Berger, what did they find?

Line 11-12: This sentence is confusing, I am unclear if they are still talking about causes of the MBT or if they are now talking about drivers of vegetation change

Line 23: This sentence doesn't quite make sense as is.

Line 28: Castaneda and Johnson refs, These two are not pollen records, as stated above. Should also add Bosumtwi pollen record maybe.

*Page 3*

Line 23: impacts on Southern Hemisphere, why is this?

Line 30: what are the bergwinds and how do we know that they don't transport much materials?

*Page 5*

Line 25: Is this referring to $\delta D_{wax}$?

*Page 6*

Line 7: represented, not presented

Lines 17-19: low sediment transport could also be because of less erosion with denser vegetation and root networks, rather than drier conditions.  Maybe some discussion of seasonality, particularly as regards the expansion of woodland could be useful.

*Page 8*

Lines 11-12: what is the evidence for this?

Line 15: elemental not element

Line 27: what is the physiological mechanism here?  miombo is more drought adapted, you would think the opposite might be true?

*Page 9*

Line 31:  development of what?

---

## Author Comment (AC1) · 25 Apr 2019

We thank the reviewer for the thoughtful comments helping us to improve the manuscript substantially. In the following, we respond to each specific point.

The paper reports on a prominent and unique pollen record from an Indian Ocean core and provides interesting comparisons with relevant other records derived from different sources including proxies from the same core and others like the Vostok Ice core. The paper is in connection with the influence atmospheric CO2 concentrations on eastern Southern African vegetation types and suggests that under the higher concentrations of pCO2 woody plant growth thrives. I can't say I am entirely convinced by the cause and effect arguments that are proposed between pCO2 and other factors like hyrdroclimate and temperature as is acknowledged by the authors, and therefore would suggest a more cautious approach.

We have revised the section about the effects of $CO_2$ and deepened the discussion. The section now reads as follows:

*Effects of atmospheric $pCO_2$*

[revised manuscript text omitted]

Proxies of these conditions seem to behave in concert but it is not that clear which one is the leading factor and how much they influence each another. Although the role C4 sedges seem to be important, I feel the role of C4 grasses are underplayed (discussed further below).

The grass pollen percentages vary under 20% and that is quite low compared to other marine records adjacent to deserts and savannahs. We, therefore, believe that in the biome fluctuations we do record, grasses are not that important.

This is nevertheless an important study worthy of publication provided that attention is given to the aspects listed below.

Abstract

P1, line 16. If possible, please mention to what degree and on what basis the effects of the factors, hydroclimate, temperature and atmospheric pCO2, can be disentangled.

The disentanglement is rather complex. We argue that (1) the precessional rhythms of river discharge compared to the interglacial-glacial biome variability indicates that hydroclimate cannot be the only driver of vegetation change. The other options of forcing mechanisms on interglacial-glacial time-scales are temperature and $p\text{CO}_2$. (2) Because of the correlation between Cyperaceae pollen percentages and $\delta^{13}\text{C}_{wax}$ and the lack of correlation between Poaceae percentages and $\delta^{13}\text{C}_{wax}$ in combination with the relatively low grass pollen percentages, we deduce that the C4 plant imprint mainly derives from the sedges. (3) The expansion of C4 sedges during the colder periods of the glacials is unlikely to result from lower temperatures. (4) The confinement of mountain forest to the glacial periods with moderate temperatures and moderate $p\text{CO}_2$, and the lack of extension into the lowlands of mountain forest during the colder periods suggest that low $p\text{CO}_2$ became restrictive to the forest. (5) Based on the elemental composition as a proxy for river discharge, we estimate the post-MBE interglacials as the drier intervals of the sequence. Nevertheless woodland expanded during those periods, which we attribute to increased temperatures and $p\text{CO}_2$.

We feel reluctant to put this rather lengthy list into the Abstract but we incorporate it in the Conclusions section.

P1, lines 17, 18, 19. The statement could provide better insight if it can be more specific, e.g., do these different vegetation categories respond in the same way or differently to pCO2? The word "depended" might have to be reconsidered in view of the above concerns. Please insert (_430 ka) after Mid-Brunhes Event.

We change this part of the Abstract into:

*Our results suggest that the extension of mountain forest occurred during those parts of the glacials when pCO₂ and temperatures were moderate and that only during the colder periods when atmospheric pCO₂ was low (less than 220 ppmv) open ericaceous vegetation including C4 sedges expanded. The main development of woodlands in the area took place after the Mid-Brunhes Event (~430 ka) when interglacial pCO₂ levels regularly rose over 270 ppmv.*

Introduction

P1, line 24. If subscript is used in pCO2, why not in C4?

If allowed, we'd like to reserve the use of subscripts for the chemical/physical terminology.

P2, line 13. It is unclear what kind of event is meant here.

We rewrite that part of the paragraph as follows:

*The climate transition of the MBE has been extensively studied using Earth System Models of Intermediate Complexity. Yin & Berger [2010] stress the importance of forcing by austral summer insolation and Yin & Berger [2012] argue that the model vegetation (tree-fraction) was forced by*

*precession through precipitation at low latitudes. Both papers show the necessity to include the change in atmospheric $CO_2$ in the explanation of the MBE [Yin & Berger, 2010, 2012]. Yin [2013], however, concludes that it is not necessary to invoke a sudden event around 430 ka to explain the increased interglacial $CO_2$; the differences between interglacials before and after the MBE can be explained by individual responses in Southern Ocean ventilation and deep-sea temperature to various combinations of the astronomical parameters.*

P2, line 20. The phrase "to counter" could be seen as ambiguous or is it meant to be "is counter to"?

Bouttes et al. show in their model that the effects of vegetation change run in the other direction and thus counteract the effects of oceanic response. We change "counter" to "counteract".

Material and Methods

P4, line 29. Does "windows" mean sections? A question arises here why there are two older windows and not one continuous one. Is there a hiatus or another reason?

There is no hiatus; the sedimentation in the core is continuous (see Caley et al. 2018). Just for practical reasons and time constraints, I did not analyse all of the 37.6 m of sediment covering 2.2 million years at 5 ka resolution. The Early Pleistocene parts are shown for comparison with that of the Brunhes Chron, only. Therefore, we call them windows.

P5 line 11. A definition of an endmember would be helpful here to keep the uninitiated reader informed.

We adapt the part of the paragraph as follows:

*Additionally, we carried out a multivariate analysis in the form of an endmember model unmixing procedure [Weltje, 1997], the statistics of which are specifically designed for the treatment of percentage data. We regard the pollen percentages as a series of pollen assemblage mixtures, whereby each modelled endmember may be interpreted as a characteristic combination. This linear mixing model can be compared to a ternary diagram but allowing for more than three axes.*

P5, line 30. There may be correlation between sedge pollen and leaf wax isotopes but it looks as if there is also correlation with Poaceae pollen (see also below in connection with P9, line 15).

It might look so, but it is not the case. Table 1 shows that there is no significant correlation between Poaceae pollen percentages and $\delta^{13}C_{wax}$. There is a correlation between Cyperaceae pollen percentages and $\delta^{13}C_{wax}$ and despite the correlation between Poaceae and Cyperaceae pollen percentages there is no correlation between Poaceae pollen percentages and $\delta^{13}C_{wax}$.

P6, lines 6,7. Why is this relevant? Won't one find C4 sedges near any of these African lakes?

The sentence "Thus, fluvial discharge was probably low during interglacials …" is the conclusion of the paragraph that discusses and refutes the possibility of sea-level changes driving the variability seen in Cyperaceae pollen values and Fe/Ca ratios.

P6, lines 19,20. I can't see why this is remarkable. In my experience dry conditions result in less ground cover hence relatively more woody elements.

We drop the phrase: "It is remarkable that". However, we do not think we see encroaching of woody vegetation in a grassy environment. Less ground cover and encroaching would have occurred under much drier conditions, in which case we would have seen a decrease of relatively high Poaceae pollen percentages and an increase of pollen percentages representative for shrubland and desert. This is not the case for the post-MBE interglacials when E-woodland shows maximum values. We see

a change from E-Heathland to E-Woodland at the last glacial-interglacial transition and a change from E-Mountain-Forest to E-Woodland in the transitions to interglacial conditions of MIS 11c, 9e, 7e, and 5e. Only during interglacial 7a, a peak of E-shrubland is recorded.

Endmembers representing vegetation on land

P6, line 23. In connection with the three endmembers, the reader would by now have seen from Figure 3 that there are 4 endmembers. Therefore, I suggest saying "characterized initially". The word "so-called" should have been used when endmembers were first mentioned. A table in the text with the most prominent constituents of the endmembers will make it easier to understand the significance without having to go and look in the supplements.

Thank you for the suggestion. We insert "initially" and drop "so-called". A clarification of the terminology endmember is now given in the Material and Methods section.

However, we refrain from adding a table to the main text. The supplementary tables are far too big to include in the text. A selection would do no more than illustrate the text of the section "Endmembers representing vegetation on land". Only the full data set of the supplementary tables allows the reader to check whether we made a reasonable selection.

P7, line 2. Must be "endmember's"?

Yes, done.

P7, line 3. This "one endmember" is a little confusing. Say which one or say: one endmember had a counterpart in EM2.

Thank you; we change the text according your suggestion.

P7. Line 15. What kind of extremes?

We change the sentence into: "The fractional abundance of the E-Mountain-Forest is also high in glacials of the Brunhes Chron but not during the extreme glacial stages, when temperatures and $p$CO$_2$ are particularly low (Figure 3)."

P7, line 30. "wide range woodland taxa" like Combretaceae? I see this important taxon in not mentioned except in the supplements.

Yes, Combretaceae fit into E-Woodland. We add "such as Combretaceae species" after "wide-range woodland taxa".

P8, line 8. This may be seen as ambiguous if "developed" is taken as originated. Did miombo not start much earlier? Effects of atmospheric pCO2

Sure, Miombo woodland would have originated outside the region. We change the sentence into: "It is likely that the Miombo dry forest and woodland migrated into the region in the successive interglacials of the Brunhes Chron."

P9, line 15. Surely some C4 grasses also thrive under relatively moist conditions if it is not too cold. Seasonality or growing season moisture is a factor which was not evaluated enough in the paper. Is it not possible that apart from C4 sedges, certain C4 Poaceae also played a role? On high land, frost and winter seasonality might be a factor ruling out C4 grasses in favour of C3 grasses or small shrubs, but as long as there are summer rains and subtropical Africa is warm enough during the glacial periods, which will probably be the case on the low-lying coastal platform, C4 grasses will be supported. It might be worthwhile to consider Vogel's work on the distribution of C3/C4 grasses in Southern Africa in this study. There are also some arguments in this connection in Scott (2002).

It is somewhat speculative to deduct changes in seasonality from our data. Most taxa comprising E-Woodland are adapted to seasonal climates with summer rainfall. Changes from mountain forest to woodland might suggest increase in seasonality. However, increase in seasonality would not decrease river discharge. Many elements of E-Heathland nowadays grow under winter rainfall. However, to propose a winterrain climate as far north as Mozambique during glacials is unrealistic and not supported by other paleodata or modelling studies.

Our Poaceae pollen percentages are relatively low and do not fluctuate with the $\delta^{13}C_{wax}$. We do not doubt that there have been C4 grasses but do not think that C4 grasses played a major role in the fluctuations we see. We add "We also note that in many parts of South Africa, no substantial increase of C4 grasses occurred but that many sites suggest an expansion of C3 grasses during the Last Glacial Maximum [Scott 2002]." at the end of the paragraph starting with "Pollen records of ericaceous vegetation suggest an extensive open vegetation existing in the East African Mountains…" (see also the revised text of the section "Effects of atmospheric $p$CO$_2$", above).

The revised manuscript showing all changes is uploaded as supplement.

On behalf of Thibaut Caley and Isla Castañeda

Lydie Dupont

Vogel, J.C., Fuls, A., Ellis, R.P., 1978. The geographical distribution of Kranz grasses in South Africa. S. Afr. J. Sci. 74, 209- 215.

Scott, L. 2002. Grassland development under glacial and interglacial conditions in Southern Africa: review of pollen, phytolith and isotope evidence. Palaeogeography, Palaeoclimatology, Palaeoecology 177(1-2): 47-57.

---

## Author Comment (AC2) · 25 Apr 2019

Response to reviewer #2, Sarah Ivory.

We thank Sarah Ivory (anonymous reviewer #2) for the thoughtful comments helping us to improve the manuscript substantially. In the following, we respond to each specific point.

The manuscript by Dupont et al. uses pollen assemblages and other multi-indicator data from a marine sediment core in order to look at vegetation and climate change in southeastern Africa over multiple glacial-interglacial cycles. They extend a previously published dataset to 800k and also include analysis of older mid and early Pleistocene sections in order to disentangle the effects of temperature, precipitation, and $pCO_2$ on vegetation.
I find this paper to present a very well done and important dataset from a data-sparse region of the tropics. I am really happy to see the pollen analyses extended to older sections of the core and appreciate the data-rich multi-indicator approach. I also support the end-member analysis conducted by the authors, which seems to be able to be used to tease apart assemblages through time. I also really appreciate the comparison of the carbon isotope data with pollen data, which I find enhances the interpretations from both datasets.
I do however have a few concerns about the paper, these are mostly minor. In particular, I find that the main conclusions about $pCO_2$ and its impact on vegetation over time is a bit thinly supported. No further data analysis supports this argument and in fact I find that the section which describes the patterns attributed to $CO_2$ is very brief given the prominence of this driver in the abstract of the paper. My most pressing suggestion for improvement of this paper then is to further develop this section of the paper and perhaps include a plot showing major vegetation types under differing $CO_2$ thresholds discussed. The wiggle plots which represent the bulk of this argument in Figures 4 and 5 are not sufficient for really teasing apart the impact of $CO_2$ or illustrating the authors' interpretation.

We have revised our discussion about the effects of $CO_2$ as follows:

*Effects of atmospheric $pCO_2$*

[revised manuscript text omitted]

We added frequency analysis of the XRF ln(Fe/Ca) and Cyperaceae pollen percentages in the Supplementary Information (trailing this response).

We also adapted Figures 4 and 5 to highlight our interpretation.

[Figure]

Figure 4: Comparing pollen assembages E-Mountain-Forest, E-Woodland and E-Shrubland with atmospheric $CO_2$ [Bereiter et al. 2015, PAGES 2016]. On top Interglacial peaks of the past 800 ka [PAGES 2016] and stable oxygen isotopes of benthic foraminifera ($\delta^{18}O_{benthic}$) of MD96-2048 [Caley et al. 2011]. $CO_2$-levels of 220 and 270 ppmv are indicated with green-red shading. Grey shading highlight periods with maximum atmospheric $CO_2$ and maximum values of the sum of E-Woodland and E-Mountain-Forest. VPDB: Vienna Pee Dee Belemnite.

[Figure]

Figure 5: Comparing glacial pollen assembages E-Mountain-Forest and E-Heathland with atmospheric $CO_2$ [Bereiter et al. 2015, PAGES 2016] and sea surface temperatures of the southeatern Indian Ocean (SST PC1 scores of MD96-20468) [Caley et al. 2018]. On top Terminations of the past 9 glacacions and stable oxygen isotopes of benthic foraminifera ($\delta^{18}O_{benthic}$) of MD96-2048 [Caley et al. 2011]. $CO_2$-levels of 220 and 270 ppmv are indicated with blue-green-red shading. Grey shading highlight periods with minimum atmospheric $CO_2$, minimum values of E-Mountain-Forest, and maximum values of E-Heathland. VPDB: Vienna Pee Dee Belemnite.

*Page 1*
-Please switch to sequential line numbering for the whole paper, rather than per page, if that fits with journal formatting.

It is the CP-format…

Line 26: I think this isn't quite accurate, because C4 vegetation is by its very nature arid adapted in that C4 metabolism is an adaptation to increase water use efficiency. I think instead you should change this sentence to imply that the risk misinterpreting the bioclimatic controls on the expansion of arid adapted vegetation.

We do not agree that all C4 vegetation is by its very nature arid adapted. In C4 grasses the C4 metabolism may be an adaptation to water use efficiency but certainly not in *Cyperus* species that grow in wetlands. Stock et al. (2004) write in their introduction concerning C4 photosynthesis in South African Cyperaceae: "that C4 photosynthesis probably evolved under wetland conditions for species of the genus *Cyperus* and that the ecological success of the group in infertile wetlands is a consequence of the high nitrogen use efficiency associated with the C4 pathway." see also Li et al. (1999).

Nevertheless, we agree that the sentence "Comparing this open C4 rich vegetation with modern analogues would have led to an interpretation of the occurrence of arid adapted vegetation."

should be rephrased, which we do as follows:

*Comparing records of this glacial C4-rich vegetation with modern analogues could have led to estimating more severe aridity than actually occurred during the Last Glacial Maximum.*

-Which studies?

The studies cited before. We insert [opt cit.]

Also this phrasing is awkward, I think "last Glacial-interglacial transition" or Pleistocene-Holocene transition might be better

Yes, thank you. We change "Glacial Holocene" to "last glacial-interglacial transition"

Line 27: "Glacial" glacials?

No, we mean Early Glacial: MIS 5a-d. We should have capitalized 'Early'. We adapt the text accordingly.

Line 4: "cycle" should be cycles

Done

Line 11: Yin and Berger, what did they find?

Line 11-12: This sentence is confusing, I am unclear if they are still talking about causes of the MBT or if they are now talking about drivers of vegetation change

We rephrase as follows:

*The climate transition of the MBE has been extensively studied using Earth System Models of Intermediate Complexity. Yin & Berger [2010] stress the importance of forcing by austral summer insolation and Yin & Berger [2012] argue that the model vegetation (tree-fraction) was forced by precession through precipitation at low latitudes. Both papers show the necessity to include the change in atmospheric $CO_2$ in the explanation of the MBE [Yin & Berger, 2010, 2012]. Yin [2013], however, concludes that it is not necessary to invoke a sudden event around 430 ka to explain the increased interglacial $CO_2$; the differences between interglacials before and after the MBE can be explained by individual responses in Southern Ocean ventilation and deep-sea temperature to various combinations of the astronomical parameters. On the other hand, statistical analysis suggests a dominant role of the carbon cycle, which changed over the MBE [Barth et al. 2018].*

Line 23: This sentence doesn't quite make sense as is.

Yes, we drop the second half of the sentence, which now reads: "Comparing records of pre- and post-MBE interglacials could offer insight in the interglacial climate at different levels of $CO_2$ [Foley et al. 1994, Swan et al. 2010]."

Line 28: Castaneda and Johnson refs, These two are not pollen records, as stated above. Should also add Bosumtwi pollen record maybe.

We change pollen records into vegetation records. We add the Bosumtwi record as suggested. However, we dropped the El'gygytgyn record because we realized that no interglacials of the early Brunhes Chron were analyzed in detail. We add the ODP Site 658 record instead.

The paragraph reads now as follows:

*Comparing records of pre- and post-MBE interglacials could offer insight in the interglacial climate at different levels of CO2 [Foley et al. 1994, Swan et al. 2010]. We define interglacials after PAGES [2016] listing MIS 19c, 17c, 15a, 15e, 13a as pre-MBE and MIS 11c, 9e, 7e, 7a-c, 5e, 1 as post-MBE. Currently, only a handful of vegetation records covering the entire Brunhes Chron have sufficient temporal resolution to enable comparisons between interglacials before and after the Mid-Brunhes transition. These records are from the eastern Mediterranean, the Colombian Andes [PAGES 2016], West and East Africa [Dupont et al. 1989, Miller & Goslin 2014, Castañeda et al. 2016, Johnson et al. 2016, Ivory et al. 2018, Owen et al. 2018]. The Andean pollen record is strongly influenced by the immigration of oak from North America during MIS 12 [Torres et al. 2013]. For the eastern Mediterranean a decline in plant diversity is observed at Tenaghi Phillipon (Greece) where the modern Mediterranean oak forests gradual emerged in the interglacials after MIS 16 but before the MBE [Tzedakis et al. 2006, 2009]. The West African record of Lake Bosumtwi in Ghana allows identification of six forest assemblages since 540 ka related to the interglacials of MIS 13, 11, 9, 7, 5e, and 1. The forests assemblage of MIS 13, however, does not show a strong contrast with those of the interglacials after the MBE [Miller & Goslin 2014]. The marine pollen record of ODP Site 658 off Cape Blanc tracks the latitudinal position of the open grass-rich vegetation zones at the boundary between Sahara and Sahel suggesting shifting vegetation zones between glacials and interglacials [Dupont & Hooghiemstra 1989, Dupont et al. 1989]. The drier interglacials occurred after MIS 9, which indicates a transition after the MBE to more arid conditions. Additionally, stable carbon isotope records from Chinese loess sections indicate interglacial-glacial variability in the C3-C4 proportions of the vegetation [Lyu et al. 2018, Sun et al. 2019]. However, the latter records do not show a prominent vegetation shift over the MBE.*

Line 23: impacts on Southern Hemisphere, why is this?

According to the modelling study of Yin & Berger (2010), the seasonal distribution of insolation is such that the Southern Hemisphere receives more summer insolation during the post-MBE interglacials which enhances the effect of higher $p\mathrm{CO_2}$. We change the sentence as follows:

*We might expect a change of Southern Hemisphere vegetation being less ambiguous than the changes found on the Northern Hemisphere (see above), because modelling indicates that the effects of the MBE were more pronounced on the Southern Hemisphere [Yin & Berger 2010].*

Line 30: what are the bergwinds and how do we know that they don't transport much materials?

The bergwinds might transport some material from the northern part of the Drakensberg, which would bring pollen from the same area as the rivers do draining the escarpment and the Lembombo Hills.

Line 25: Is this referring to δDwax?

Yes, but as the terminology turns up only twice and only in referring to another study, we refrain from introducing the abbreviation.

Line 7: represented, not presented

Done

Lines 17-19: low sediment transport could also be because of less erosion with denser vegetation and root networks, rather than drier conditions. Maybe some discussion of seasonality, particularly as regards the expansion of woodland could be useful.

We doubt that a change from mountain forest to woodland or from heathland to woodland would have decreased erosion. We have little indication of biomes that imbed erodible bare soils with the possible exception of E-Shrubland, which has the higher values in the early Pleistocene.

Most taxa comprising E-Woodland are adapted to seasonal climates with summer rainfall. Changes from mountain forest to woodland might suggest increase in seasonality. However, increase in seasonality would not decrease river discharge. Many elements of E-Heathland nowadays grow under winter rainfall. However, to propose a winterrain climate as far north as Mozambique during glacials is unrealistic and not supported by other paleodata or modelling studies.

Lines 11-12: what is the evidence for this?

We want to draw attention to the different paces of vegetation and river discharge as shown by the fluctuations of E-Heathland, in particular. We added in the Supplementary Information spectral analyses of XRF ln(Fe/Ca), E-Heathland fractional abundances, and Cyperaceae pollen concentration (see also above).

Line 15: elemental not element

Done

Line 27: what is the physiological mechanism here? miombo is more drought adapted, you would think the opposite might be true?

We do think that Miombo is better drought adapted than the mountain forest or the coastal forest and probably also better drought adapted than a fynbos-like vegetation. However, we don't understand the question because the hypothesis is about the effect of enhanced $p$CO$_2$ during interglacials.

Line 31: development of what?

We mean Lake Magadi. We change the beginning the the prargraph as follows:

*The trend to increased woodland in SE Africa after the MBE, noted at both Lake Malawi and in the Limpopo River catchment [Johnson et al. 2016, Caley et al. 2018, this study] contrasts with the trend*

*around Lake Magadi at the equator. At Lake Magadi a trend to less forest around marks the Mid-Brunhes transition [Owen et al. 2018].*

The revised manuscript showing all changes is uploaded as supplement.

On behalf of Thibaut Caley and Isla Castañeda

Lydie Dupont

ADDITIONAL REFERENCES

Li M.-R., Wedin D. A., Tieszen L. L., 1999. C3 and C4 photosynthesis in Cyperus (Cyperaceae) in temperate eastern North America. Can J. Bot. 77, 209-18.

Dupont, L.M. & Hooghiemstra, H., 1989. The Saharan-Sahelian boundary during the Brunhes chron. Acta Botanica Neerlandica, 38: 405-415.

Dupont, L.M., Beug, H-J., Stalling, H. & Tiedemann, R., 1989. First palynological results from ODP Site 658 at 21°N west off Africa: pollen as climate indicators. In: Ruddiman, W.F., Sarnthein, M. et al. Proceedings ODP Scientific Results, 108, College Station TX (Ocean Drilling Program): 93-111.

Miller, S.M. & Gosling, W.D., 2014. Quaternary forest associations in lowland tropical West Africa. Quaternary Science Reviews, 84: 7-25.

Simon, M.H., Ziegler, M., Bosmans, J., Barker, S., Reason, C.J.C. & Hall, I.R., 2015. Eastern South African hydroclimate over the past 270,000 years. Scientific Reports, 5, 18153: 1-10.

Singarayer, J.S. & Burrough, S.L., 2015. Interhemispheric dynamics of the African rainbelt during the late Quaternary. Quaternary Science Reviews, 124: 48-67.

Wu, H., Guiot, J., Brewer, S. & Guo, Z., 2007. Climatic changes in Eurasia and Africa at the last glacial maximum and mid-Holocene: reconstruction from pollen data using inverse vegetation modelling. Climate Dynamics, 29: 211-229.

**Supplementary Information**

REDFIT frequency analysis

We conducted a frequency analysis on the data of the ln(Fe/Ca) ratios, the E-Heathland fractional abundance scores, and the Cyperaceae pollen concentration covering the Brunhes Chron using the algorithm of REDFIT [Schulz & Mudelsee 2002] from the statistical package PAST version 3.14 (1999-2006) [Hammer et al. 2001]. The E-Heathland and Cyperaceae curves each consisted of 181 data points between 0 and 790 ka. REDFIT was run with 2 times oversampling, a Blackman-Harris window, and 2 overlapping averaging segments resulting in a bandwidth of 0.004291; false alarm level was 99.17. The ln(Fe/Ca) curve contained 2307 data points between 1 and 790 ka. REDFIT was run with 2 times oversampling, a Blackman-Harris window, and 3 overlapping averaging segments resulting in a bandwidth of 0.005726; false alarm level was 99.91. The figure shows the power of ln(Fe/Ca) ratios (left), the power of the E-Heathland values (middle), and the power of the Cyperaceae pollen concentration (right) against frequency running from 0 - 0.08 cycles per ka. Denoted are the bandwidth for each spectrum and a parametric approximation of the level above the null hypothesis of a red noise model using $X^2$-test at 90% (dashed lines). Grey bars indicate the orbital periodicities of 100, 41, 23, and 19 ka). Note the maximum in spectral density at 23 ka (precession) in the power spectrum of ln(Fe/Ca) and the lack of spectral density at the precession bands (23 and 19 ka) in the power spectrum of the E-Heathland values. The Cyperaceae pollen concentration, which is both influenced by the expansion of Cyperaceae (sedges) and by the transport of pollen by river discharge, shows significant power at both the 100 and 19 ka.

---

## Author Response (AR1)

**Editor Decision: Publish subject to minor revisions (review by editor)** (07 May 2019)

Thank you for yours responses to reviewers. As you saw, the two reviews are positive and conclude on minor revisions. I have carefully read your responses to the reviewers and your propositions to modify the text of your manuscript.

Your paper is very interesting and I appreciate your detailed responses. Nevertheless, your manuscript needs some additional amendments corresponding to some of the comments of the two reviewers and few questions for which I would like your responses.

1- Following the comments of reviewers 2, I think that Bergwinds have to be explained to readers in the text that may be interested on this paper but not familiarized on the studied area.

We insert: "offshore winds descending from the interior plateau" (Page 5, Line 6-7)

2- I have some problems with definitions of endmembers as reviewer 1. Probably it will be easier to understand if when describing the endmembers in the corresponding paragraph you separate the different endmembers and their composition by subtitles each one for an endmember and then the correspondence with the previous one. For my part I am a bit lost in this part.

Each paragraph concerns 1 endmember, which we mention at the beginning of the paragraph (Page 9, Line 14, Page 10, Lines 8, 16, 32. We also introduce these names in the preceding paragraph (Page 9, Line 1-13).

In fact I really think as the reviewer 1 that a short table in the main text will be helpful even if we have some details in the supplement with the curves of all taxa presented separately (do the curves of all taxa be in the supplementary figures?).

OK, we add Table 2 (Page 10)

Table 2. Interpretation of the endmembers

| Endmember | Main pollen taxa |
|---|---|
| E-Heathland | *Podocarpus, Celtis, Olea* |
| E-Mountain-Forest | Cyperaceae, Ericaceae, *Phaeoceros*, Restionaceae, *Stoebe* type, *Anthoceros, Typha Lycopodium*, Restionaceae |
| E-Shrubland | Poaceae, Asteroideae, *Buxus*, Amaranthaceae, *Euphorbia*, Meliaceae-Sapotaceae, *Acacia, Riccia* type, *Tribulus*, Acanthaceae pp, Asteraceae Vernoniae, *Hypoestes-Dicliptera* type, *Gazania* type, *Dombeya* |
| E-Woodland | *Alchornea, Spirostachys africana, Pteridium* type, Polypodiaceae, *Myrsine africana, Cassia* type, Rhizophoraceae, *Aizoaceae*, Combretaceae pp, *Manilkara, Burkea africana, Brachystegia, Dodonaea viscosa, Pseudolachnostylis, Hymenocardia, Aloe*, Rhamnaceae pp, *Protea, Parinari* |

All curves mentioned in the text are plotted in the supplementary figures. I consider putting the supplementary figures into the main text, but that might be overkill. What do you think?

In fact you write the term endmembers in page before define it as the reviewer 1 said. what you add in the matérial and method sections is very short. what do you mean with "characteristic combination?

Something that can be interpreted. We rephrase as follows:

"We regard the pollen percentages as a series of pollen assemblage mixtures, whereby each modelled endmember may be interpreted as the representation of one or more biomes." (Page 6, Line 30 - Page 7, Line 1)

You add a citation of table 1 and table 2 supplementary documents but they are not in the documents given with the responses, I did not find them.
Please explain me what happened and /or produce these tables.

Supplementary Tables 1 and 2 have not changed. I upload them again. The supplementary figures are trailing the manuscript.

3- Concerning your corrected discussion I have some remarks:
a. In the figures it will be better to indicate glacial and interglacial numbers, sometimes you mark the glacial numbers sometimes the interglacial ones sometimes the terminations. I did not understand the choice (numbering glacial and interglacial or terminations???) in the different figures even in the supplementary ones. Explain me.

The choice for Terminations is that they clearly separate the climate cycles. We denote interglacial stages in the figures showing curves of vegetation variability associated with interglacials, such as E-woodland and E-shrubland. We use Terminations in the figures denoting vegetation variability associated with glacials, such as E-Heathland and A-Mountain-Forest. We add even MIS numbers at Figure 5.

b. End of p. 11 (new text) you say that mountain forest is replaced by woodlands in 11c, 9e, 7e, 7c, 5e and 1. First will you mark the in the right order. I do not see that in 7e and 7c. I see only a slight regression of mountain taxa and a slight increase in woodland in these two events.

Yes, 'replaced' is not the right word. We change the sentence into: "Woodlands would have expanded at the cost of mountain forest during 11c, 9e, 5e and 1, and to a lesser extend during 7e and 7c, " (Page 12, Line 7-10)

c. p. 12 first sentence you explain "the record……during those parts of glacial stages". I do not understand, you were talking about interglacials???? And then the term "those part of the glacials stages" refers to what? I am lost here. Why did you talk about glacial in the description of interglacials?

Actually, the biomes and not the glacials or interglacials are the main themes of those paragraphs. We rearranged the paragraph about mountain forest as follows (Page 12, Line 12-16, Page 13, Line 1-2):

The glacial stages showed the expansion of either mountain forest or heathland. The record indicates extension of mountain forests in SE Africa during those parts of the glacial stages with low temperatures and atmospheric $pCO_2$ exceeding ~220 ppmv (Figure 5).  If low temperatures were the only driver of the extension of mountain forests, further spread into the lowlands during the coldest glacial phases should be expected. Instead, when $pCO_2$ dropped below ~220 ppmv during those colder glacial periods, mountain forest declined, in particular during MIS 18, 16, 14, 8, 6, and 2. A picture emerges of cool glacial stages in SE Africa in which tree cover broke down when atmospheric $pCO_2$ became too low. Additionally, mountain forests were important during the Interglacials 19c, 17c, 15e, 15a, 13a, and 7e, in which $pCO_2$ and Antarctic temperatures were subdued.

You talk about Podocapus and you cited the fig. 5 why? Podocarpus is not on the figure 5 only in additional figures.

We changed Podocarpus forest into mountain forest (see above).

d. Line 4 p.12 you explain that you have mountain forest in the coldest glacial just before the interglacials marked by the replacement between the two types of vegetation so MIS 18, 16, 14, 8, 6 and 2. First, here fig. 5 has to be cited please and it will be helpful to write at least the MIS glacial numbers on the figure).

Figure 5 (Page 1, Line 15) is cited and we add MIS glacial stages to the figure.

Have you an explanation for the difference observed in the 12 and 10? They are not so different from the 14.

No, not really.

e. I need also some links between the C4 vegetation, the C4 sedges and the endmembers in the text. In the beginning of the discussion, I understood that the C4 vegetation is represented in you record mainly by sedges. Is it true?

Yes

Then it mainly corresponds to old EM2 and new E heathland? Is it true?

Yes, see discussion section about endmember interpretation

Please add this in some sentences of the discussion to be clear.

We add sedges percentages (Cyperaceae %%) to Figure 5. We also insert "being an important constituent of the ericaceous fynbos-like vegetation" after C4 sedges (Page 13, Line 12-13) and "(see also the correlation between SST and EM2 in Dupont et al. 2011)" after correlating with the SST record (Page 13, Line 34).

Please clarify these points in the discussion and organize the discussion in a different way to be more clear for the reader (first the interglacials and secondly the glacials) and cite the supplementary figures if necessary.
I do not think that these last remarks and comments take a long time to be addressed. So I am waiting after your new text and then will be able to post my final decision on the publication of this very interesting manuscript

With my best regards
Nathalie Combourieu-Nebout

Thank you very much. I hope the changes and responses are satisfactory.

With kind regards and greetings,
Lydie Dupont

[revised manuscript text omitted]